

# Invasive success of star weed (*Parthenium hysterophorus* L.) through alteration in structural and functional peculiarities

Ummar Iqbal[1,*], Zartasha Usman[1], Akkasha Azam[1], Hina Abbas[1], Ansar Mehmood[2] and Khawaja Shafique Ahmad[2,*]

[1] Botany, The Islamia University of Bahawalpur, Rahim Yar Khan Campus, Rahim Yar Khan, Punjab, Pakistan
[2] Botany, University of Poonch Rawalakot, Rawalakot, Azad Jammu and Kashmir, Pakistan
* These authors contributed equally to this work.

Corresponding authors
Ummar Iqbal,
ummariqbal@yahoo.com
Khawaja Shafique Ahmad,
ahmadks@upr.edu.pk

## ABSTRACT

Parthenium weed poses significant threats to cropping systems, socioeconomic structures, and native ecosystems. The pronounced impact is primarily attributed to its rapid and efficient invasion mechanism. Despite that the detrimental effects of Parthenium weed are widely acknowledged, an in-depth scientific comprehension of its invasion mechanism, particularly regarding modifications in structural and functional attributes under natural conditions, is still lacking. To bridge this knowledge gap and formulate effective strategies for alleviating the adverse consequences of Parthenium weed, a study was conducted in the more cultivated and densely populated areas of Punjab, Pakistan. This study was focused on fifteen distinct populations of the star weed (*Parthenium hysterophorus* L.) to investigate the factors contributing to its widespread distribution in diverse environmental conditions. The results revealed significant variations in growth performance, physiological traits, and internal structures among populations from different habitats. The populations from wastelands exhibited superior growth, with higher accumulation of soluble proteins (TSP) and chlorophyll content (Chl *a*&*b*, TChl, Car, and Chl a/b). These populations displayed increased root and stem area, storage parenchyma, vascular bundle area, metaxylem area, and phloem area. Significant leaf modifications included thicker leaves, sclarification around vascular bundles, and widened metaxylem vessels. Roadside populations possessed larger leaf area, enhanced antioxidant activity, increased thickness of leaves in terms of midrib and lamina, and a higher cortical proportion. Populations found in agricultural fields depicted enhanced shoot biomass production, higher levels of chlorophyll b, and an increased total chlorophyll/carotenoid ratio. Additionally, they exhibited increased phloem area in their roots, stems, and leaves, with a thick epidermis only in the stem. All these outcomes of the study revealed explicit structural and functional modifications among *P. hysterophorus* populations collected from different habitats. These variations were attributed to the environmental variability and could contribute to the widespread distribution of this species. Notably, these findings hold practical significance for agronomists and ecologists, offering valuable insights for the future management of Parthenium weed in novel environments and contributing to the stability of ecosystems.

# INTRODUCTION

Climate change poses an existential threat to global food security, ecosystems, and public health, with atmospheric carbon dioxide ($CO_2$) levels projected to exceed 700 ppm by the century's end and average temperatures increasing by 4 °C. These changes are expected to enhance the growth and reproductive capabilities of many weeds and invasive plants, enabling them to compete more effectively with crops and pastures (*IPCC, 2014*; *Mao, Bajwa & Adkins, 2021*). Moreover, climate change may reduce the efficacy of chemical herbicides and biological control agents. In addition to long-term climate shifts, the increased frequency and intensity of extreme weather events, such as floods and droughts, can disturb ground cover, create colonization opportunities, and facilitate weed dispersal (*Sun et al., 2020*). The aggressive nature and potential impacts of parthenium weed raise concerns about the effects of climate change, particularly rising atmospheric $CO_2$ levels and temperatures, on its demography and competitive ability, as well as its management strategies. Increased temperature and reduced humidity can negatively impact biological control agents, leading to fluctuations in field population density (*Hasan & Ansari, 2016*). Furthermore, recent research focusing on elevated $CO_2$ levels suggests the need to adjust current management approaches. Therefore, a comprehensive review of parthenium weed's biology, ecology, and management options under various climate change scenarios is crucial for informing future management decisions and adapting strategies to address these emerging climate-induced challenges (*Shabbir et al., 2020*; *Mao, Bajwa & Adkins, 2021*).

In response to water scarcity and other stresses, plants adopt various survival strategies. They increase root biomass and reduce shoot growth, along with making changes in leaf orientation, size reduction, and shedding (*Leukovic et al., 2009*; *Oliveira et al., 2018*). At the anatomical level, these plants exhibit reduced cell size, enlargement in vascular tissues, alterations in the xylem/phloem ratio, and reductions in xylem and phloem vessel size (*Makbul et al., 2011*; *Boughalleb et al., 2014*). Additionally, under drought or salinity stress, plants significantly reduce xylem vessel diameter and increase the thickness of epidermis, phloem, and mesophyll tissues in aerial parts (*El Afry, El Nady & Abdelmonteleb, 2012*; *Iqbal, Hameed & Ahmad, 2023*). They also accumulate substantial amounts of protective compounds like glycine betaine, proline, and total soluble proteins to combat the adverse effects of these abiotic stresses. Ionic homeostasis is a crucial physiological mechanism in plants that contributes to their vitality and vigor even under harsh conditions (*Siringam et al., 2011*). This mechanism involves processes such as noxious ion accumulation, selective ion uptake, and excretion of toxic ions through specialized structures like leaf hairs, trichomes, leaf sheaths, and excretory organs (*Iqbal et al., 2022*).

Parthenium weed (*Parthenium hysterophorus* L.) is a highly invasive plant species that has spread across five continents, posing significant environmental, agricultural, and health threats. Originating from the neotropical region, it has rapidly expanded its range

due to accidental introductions and unchecked trade. This invasive weed has invaded diverse ecosystems, including grasslands, pastures, urban areas, and croplands, impacting biodiversity, and reducing livestock and crop production (*Adkins & Shabbir, 2014*; *Maharjan et al., 2020*). Its competitive and allelopathic effects have challenged farming systems' sustainability. Additionally, it directly endangers human and livestock health. Efforts to control and manage parthenium weed are crucial for safeguarding the environment, agriculture, and public well-being (*Shabbir et al., 2013*; *Bajwa et al., 2016*). Parthenium weed exerts deleterious impacts on various cropping systems, socioeconomic structures, and native ecosystems. The severity of these impacts is largely attributed to its swift and effective invasion mechanism, as highlighted by research (*Tanveer et al., 2015*; *Bajwa et al., 2016*). However, a comprehensive understanding of this invasion mechanism and its associated characteristics is currently lacking. To develop effective management strategies, it is imperative to gain insights into how parthenium weed invades, as well as its interactions and responses to biological and physical factors within invaded regions, which are essential for a more robust ecological comprehension.

*Parthenium hysterophorus* L., is an annual herb known for its aggressive invasion of disturbed lands and roadsides. While native to North America and Mexico, it has become an invasive species in Pakistan. This plant exhibits notable characteristics, including strong competitiveness, high drought tolerance, insensitivity to temperature fluctuations, and a remarkable capacity for seed production. Its adaptability to diverse habitats makes it an invaluable tool for studying their structural and functional responses to heterogenic environmental conditions. It was hypothesized that the invasive success of *P. hysterophorus* in diverse habitats is influenced by its phenotypic plasticity, allowing it to adapt to a wide range of environmental conditions. In this scenario, a comprehensive study was aimed to answer the following research questions: (a) How does *P. hysterophorus* respond to heterogeneous environmental conditions at the levels of growth, anatomy, and physiology? (b) What types of micro-structural, physiological, and morphological adaptations enable *P. hysterophorus* to mitigate the detrimental effects of prevailing stresses? By examining the responses and modifications at different levels, the researchers sought to gain a comprehensive understanding of the weed's ability to thrive in diverse environmental conditions.

## MATERIALS AND METHODS

### Study surveys, sampling and collection sites

Sampling was done from distinct habitats of Punjab province to determine the growth, physiological and anatomical response of *Parthenium hystyerophorus* towards heterogenic environmental conditions (Figs. 1, 2 and Table 1). Samples were collected during the peak of flowering season (March to October) in 2022. Each study site was thoroughly searched in radius of 1 km and total 50 plants were ear marked. Ten plants ($n = 10$) per population were finalized based on growth habit, plant height, shoot length, leaf number and size, flowers and inflorescence for the measurement of morpho-anatomical and physiological parameters. The populations were collected from four prominent regions such as (i) near wasteland (RYK-Rahim Yar Khan, SDK-Sadiqabad, KHP-khanpur), (ii) along water
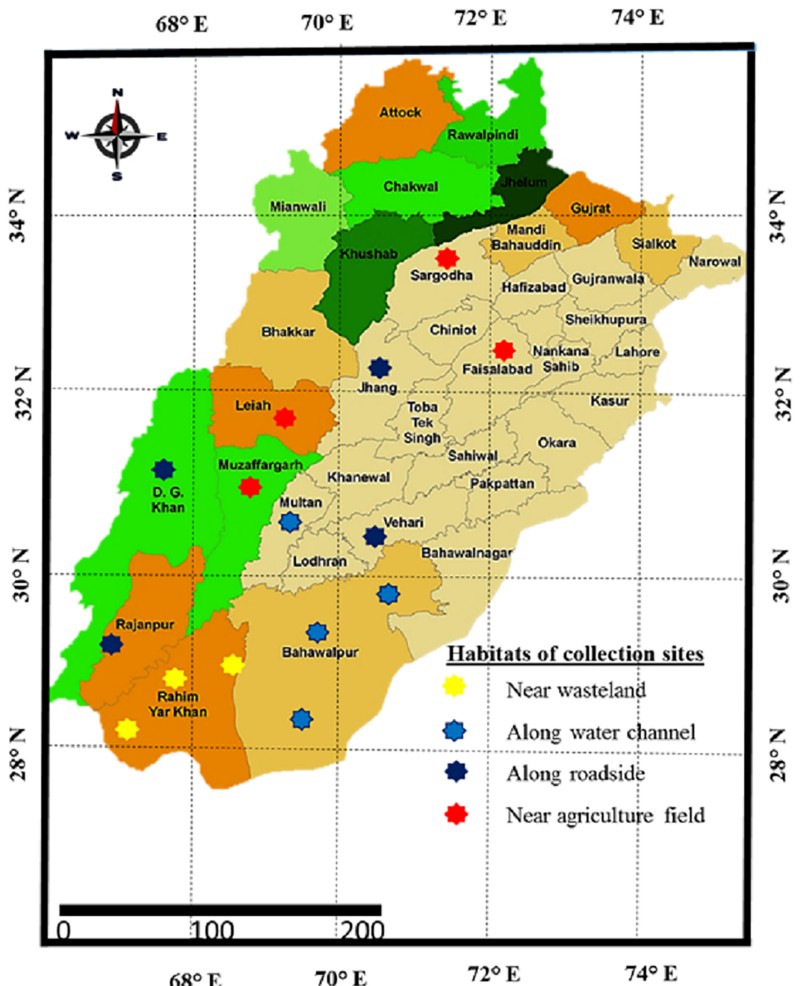

**Figure 1 Map of Punjab showing collection sites of *Parthenium hysterophorus* L. sampled from different districts.**

channels (BWP-Bahawalpur, LAP-Liaqatpur, AHP-Ahmadpur, MUL-Multan), (iii) along roadside (VEH-Vehari, DGK-DG Khan, RJP-Rajanpur, JHG-Jhang), (iv) near agriculture fields (MUZ-Muzaffargarh, SAR-Sargodha, FSD-Faisalabad, LYH-Layyah). Coordinates were measured with the help of google positioning system (GPS, model: Garmin E-Trex 20, GPS accuracy ± 1 m) (Table 2). Climatic data was taken from the meteorological department situated in each district.

## Soil physiological parameters

The soil texture was assessed using the USDA textural triangle, which categorizes soils into distinct textural classes according to the relative proportions of sand, silt, and clay present in the soil sample. The *Walkley (1947)* method was employed to measure the organic matter content (OM) in the soil. This method involves oxidation of organic matter by dichromate in the presence of sulfuric acid. A combined pH and ECe meter (WTW series InoLab pH/Cond 720) was used to measure the soil pH and electrical conductivity. Saturation paste prepared by saturating the soil with water and extracting the solution, was

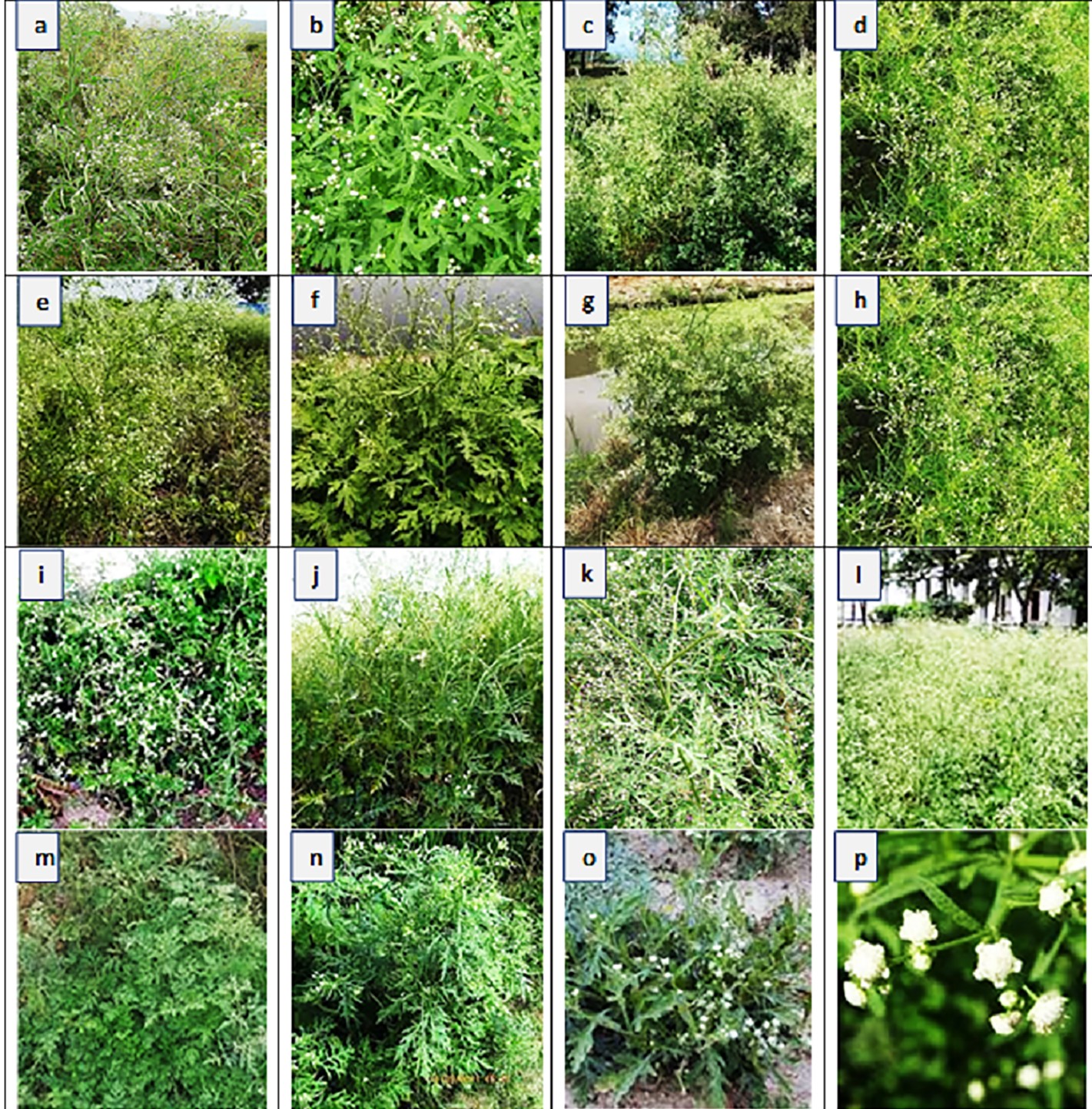

**Figure 2 Habitat view and description of *Parthenium hysterophorus* L. populations collected from different ecological regions. (A)** RYK-Rahim Yar Khan: Saline patches in the Cholistan Desert with hot and dry environment; (B) SDK-Sadiqabad: Open and barren land characterized by saline soil; (C) KHP-Khanpur: Soil is sandy, climate dry and hot. *Eucalyptus* is common genus; (D) BWP-Bahawalpur: Dominated along the bank of Indus River. Soil is sandy and hot climate, (E) LAP-Liaqatpur: Growing along the water channel in patches form, soil is sandy, (F) AHP-Ahmadpur: Vicinity of Punjab Barrage, sandy soil dominated with hydrophytes, (G) MUL-Multan: Green belt of Punjab covered by various bushes and trees like Mango, (H) VEH-Vehari: Dry and hot region characterized by loamy soil, (I) DGK-Dera Ghazi Khan: Foothills of Suleiman Mountains, climate cool in winters and very hot in summers, (J) RJP-Rajanpur: Desert flats characterized by hot climate and loamy sand. *Capparis* and *Salvadora* are

**Figure 2 (continued)**
commonly found, (K) JHG-Jhang: Sandy soil of the Thal Desert characterized by small sand dunes, climate very hot and dry, (L) MUZ-Muzaffargarh: Lush green region of the Punjab province dominated by various crops and vegetation, (M) SAR-Sargodha: Hot and dry climate having sandy soil. *Calotropis*, *Prosopis* and *Acaica* are the common species, (N) FSD-Faisalabad: It is characterized by sandy soil and hot climate, where rice and wheat are commonly cultivated crops, (O) LYH-Layyah: Flats of Thal Desert, characterized by chickpea plantation on large scale, (P) Inflorescence: The inflorescences are grouped into four or five, in small globular heads.

**Table 1  Metrological record of differently adapted populations of star weed (*Parthenium hysterophorus* L.) collected from Punjab province.**

| Ecological regions | Collection sites | Habitat types | Annual Temp. (°C) Max. | Annual Temp. (°C) Min. | Rainfall (mm) | Altitude (m.a.s.I) | Latitude (N) | Longitude (E) |
|---|---|---|---|---|---|---|---|---|
| Near wasteland | Rahim Yar Khan | Near the wasteland | 44 | 13 | 115 | 88 | 28° 42′12.29″ | 70° 29′89.19″ |
|  | Sadiqabad | Along barren land | 40 | 12 | 101 | 76 | 28° 09′19.29″ | 70° 19′12.99″ |
|  | Khanpur | Near waste deposit | 43 | 15 | 110 | 184 | 32° 08′51.27″ | 72° 38′30.22″ |
| Along water channel | Bahawalpur | Along the river Indus | 44 | 13 | 179 | 149 | 31° 08′41.23″ | 72° 08′46.38″ |
|  | Liaqatpur | Along the water canal | 34 | 14 | 119 | 237 | 32° 43′19.02″ | 72° 58′42.73″ |
|  | Ahmadpur | Near Punjab Barrage | 40 | 16 | 142 | 212 | 30° 39′31.63″ | 73° 23′50.62″ |
|  | Multan | Along Chenab River | 38 | 12 | 209 | 186 | 32° 17′43.54″ | 72° 21′03.24″ |
| Along roadside | Vehari | Near the roadside | 41 | 12 | 195 | 146 | 30° 55′46.74″ | 71° 45′41.90″ |
|  | DG Khan | Along railway track | 40 | 11 | 143 | 198 | 28° 27′42.58″ | 71° 03′919.22″ |
|  | Rajanpur | Near M5 motorway | 43 | 12 | 120 | 117 | 28° 46′04.86″ | 71° 20′03.13″ |
|  | Jhang | Near GT road | 40 | 10 | 155 | 267 | 29° 58′01.03″ | 70° 19′36.63″ |
| Near agriculture field | Muzaffargarh | Near cotton field | 44 | 13 | 176 | 210 | 32° 25′30.62″ | 37° 13′31.40″ |
|  | Sargodha | Along sorghum field | 38 | 9 | 246 | 192 | 31° 28′42.68″ | 73° 12′36.66″ |
|  | Faisalabad | Along rice field | 40 | 10 | 201 | 140 | 29° 20′05.33″ | 71° 56′04.29″ |
|  | Layyah | Wheat field | 43 | 12 | 130 | 288 | 32° 24′45.54″ | 71° 58′00.51″ |

**Table 2  Soil physicochemical parameters of collection sites of differently adapted populations of star weed (*Parthenium hysterophorus* L.) collected from Punjab province.**

| Ecological regions | Collection sites | Soil texture | ECe (dS m$^{-1}$) | pH | OM (%) | SP (%) | $PO_4^{3-}$ (mg Kg$^{-1}$) | $NO_3^-$ (mg Kg$^{-1}$) | Cl– (mg Kg$^{-1}$) | Ca$^{2+}$ (mg Kg$^{-1}$) | Na$^+$ (mg Kg$^{-1}$) | K$^+$ (mg Kg$^{-1}$) |
|---|---|---|---|---|---|---|---|---|---|---|---|---|
| Near wasteland | RYK | Loamy | 6.73[a] | 6.2[j] | 0.35[e] | 36[c] | 3.1[b] | 3.3[c] | 567.8[a] | 156.1[a] | 398.9[a] | 64.4[j] |
|  | SDK | Sandy | 0.76[h] | 8.4[g] | 0.42[d] | 16[gh] | 1.6[g] | 2.9[d] | 83.4[h] | 54.2[g] | 54.2[i] | 70.1[i] |
|  | KHP | Loamy | 6.69[a] | 8.8[b] | 0.28[g] | 38[b] | 3.4[ab] | 4.0[b] | 434.5[b] | 67.7[ef] | 297.1[b] | 260.3[b] |
| Along water channel | BWP | Sandy | 0.96[g] | 8.7[c] | 0.28[g] | 17[gh] | 1.9[d] | 2.9[d] | 102.7[g] | 71.9[e] | 147.8[f] | 80.8[g] |
|  | LAP | Sandy | 3.46[c] | 8.0[h] | 0.35[e] | 15[h] | 2.2[c] | 3.2[c] | 389.1[c] | 97.3[c] | 297.1[b] | 180.9[d] |
|  | AHP | Sandy | 1.06[f] | 8.6[d] | 0.42[d] | 16[gh] | 1.9[d] | 3.5[c] | 130.5[f] | 63.5[f] | 164.1[e] | 148.5[e] |
|  | MUL | Loamy sand | 4.33[b] | 7.8[i] | 0.56[a] | 22[d] | 2.2[c] | 3.2[c] | 72.1[j] | 60.2[f] | 266.1[c] | 276.3[a] |
| Along roadside | VEH | Loamy | 1.15[e] | 8.2[f] | 0.21[h] | 32[d] | 3.1[b] | 4.3[b] | 109.8[g] | 78.7[d] | 180.7[d] | 124.1[f] |
|  | DGK | Clayey loam | 1.19[e] | 8.7[b] | 0.28[g] | 38[b] | 3.4[ab] | 4.0[b] | 72.1[j] | 66.7[ef] | 60.8[h] | 258.3[b] |
|  | RJP | Loamy sand | 0.90[g] | 8.5[e] | 0.26[g] | 16[gh] | 1.8[d] | 2.8[d] | 100.7[g] | 70.9[e] | 145.8[f] | 79.8[g] |
|  | JHG | Loamy | 3.01[c] | 8.0[h] | 0.35[e] | 15[h] | 2.2[c] | 3.2[c] | 389.1[c] | 97.3[c] | 61.8[h] | 180.9[d] |

| Ecological regions | Collection sites | Soil texture | ECe (dS m$^{-1}$) | pH | OM (%) | SP (%) | PO$_4^{3-}$ (mg Kg$^{-1}$) | NO$_3^-$ (mg Kg$^{-1}$) | Cl– (mg Kg$^{-1}$) | Ca$^{2+}$ (mg Kg$^{-1}$) | Na$^+$ (mg Kg$^{-1}$) | K$^+$ (mg Kg$^{-1}$) |
|---|---|---|---|---|---|---|---|---|---|---|---|---|
| **Table 2 (continued)** | | | | | | | | | | | | |
| Near agriculture field | MUZ | Sandy | 1.33$^d$ | 8.2$^f$ | 0.28$^g$ | 17$^{gh}$ | 1.9$^d$ | 2.0$^e$ | 178.6$^e$ | 110.9$^b$ | 134.0$^g$ | 80.1$^g$ |
| | SAR | Sandy | 0.77$^h$ | 8.5$^e$ | 0.45$^b$ | 18$^f$ | 1.9$^d$ | 4.0$^b$ | 79.6$^i$ | 77.1$^d$ | 175.0$^d$ | 75.2$^h$ |
| | FSD | Sandy | 1.08$^f$ | 8.7$^c$ | 0.43$^{bd}$ | 19$^e$ | 2.3$^c$ | 4.0$^b$ | 111.6$^g$ | 104.3$^{bc}$ | 147.1$^f$ | 196.6$^c$ |
| | LYH | Loamy | 1.20$^{de}$ | 8.9$^a$ | 0.31$^f$ | 42$^a$ | 3.6$^a$ | 5.1$^a$ | 198.3$^d$ | 94.3$^c$ | 88.9$^g$ | 276.8$^a$ |
| | LSD | | 0.5 | 1.0 | 0.5 | 6.0 | 0.5 | 1.0 | 7.0 | 6.0 | 25.8 | 6.0 |

**Notes:**
Means shearing similar letter in each row are not statistically significant.
*Significant at $P < 0.05$, **Significant at $P < 0.01$, ***Significant at $P < 0.001$, NS, not significant.
Abbreviations are given as footnote of Figs. 7 and 8.

used for these measurements. The saturation paste was analyzed to determine the concentrations of different ions, including Na$^+$, K$^+$, and Ca$^{2+}$, utilizing a flame photometer (Jenway, PFP–7, UK). The nitrate content (NO$_3^-$) in the soil was assessed using the micro-Kjeldahl method, which involves digesting the soil sample with sulfuric acid. The resulting ammonia was then distilled and titrated using a semi-automatic ammonia distillation unit (UDK-132, NIB-B (3)-DSU-003). The soil phosphate content (PO$_4^{3-}$) was measured following the protocol described by *Wolf (1982)*. This method typically involves extracting the available phosphorus from the soil using a suitable extractant, followed by colorimetric analysis. The chloride content in the soil was assessed using the Mohrs' titration method (*Mohrs, 1856*). To determine the soil saturation percentage (SP), the soil samples were dried in an oven at 70 °C, and 200 g of the dried soil was used to prepare a composite saturation paste, which was then analyzed. Saturation percentage assayed by following formula:

$$SP\ (\%) = \frac{Amount\ of\ water\ added\ (g)}{Oven\ dried\ soil\ (g)} \times 100$$

where SP % is saturation percentage.

## Morphological parameters

To collect the necessary measurements, a meter rod was utilized to measure the plant, as well as the length of shoot and root directly. A digital loading balance was employed to determine the fresh weights of the shoot and root. Immediately after harvesting, the plant parts were weighed to obtain their fresh weights. For dry weight analysis, the plant samples were subjected to oven-drying at a temperature of 65 °C until a constant weight was achieved. This ensured the complete removal of moisture from the samples. The dry weights of the shoot and root were then measured using a digital loading balance. In the evaluation of leaf characteristics, the focus was on the uppermost mature leaves. A manual count was conducted to ascertain the quantity of leaves on each plant, and the leaf area was quantitatively measured using cm-graph paper. The leaf area was calculated using a formula provided by *Lopes et al. (2016)*.

Leaf area = Maximum leaf length $\times$ Maximum leaf width $\times$ Correction factor (0.74)

## Physiological parameters

### Osmolytes and soluble proteins

Fresh samples were taken in falcon tubes and stored (−80 °C) for chlorophyll pigments, osmoprotectants, and antioxidants activity. For the analysis of proline, fresh leaf samples were thoroughly homogenized in sulfo-salicylic acid. Then was transferred into cuvette containing ninhydrin solution. After subjected to water bath (100 °C) toluene was added for extraction of proline. Lastly, readings were taken on a spectrophotometer (Model 220; Hitachi, Tokyo, Japan) at 520 nm wavelength (*Bates, Waldren & Teare, 1973*).

$$Proline\ (\mu mol\ g^{-1} fresh\ weight) = \frac{\mu g\ proline\ ml^{-1} \times\ ml\ of\ toluene/115.5}{sample\ weight\ (g)}$$

To measure the glycine betaine content in the leaf samples, fresh leaf samples weighing 0.5 g were soaked in 20 ml of deionized water ($H_2O$) at a temperature of 25 °C for a duration of 24 h. Following the soaking period, an extract was prepared from the soaked samples and assayed using the established protocols outlined by *Grattan & Grieve (1998)*. For the analysis of total soluble proteins, fresh leaf samples weighing 0.2 g were sliced and thoroughly crushed in 5 ml of phosphate buffer at a pH of 7.0. The buffer facilitated the extraction of proteins from the crushed leaf samples. The mixture of crushed leaf samples and buffer was then subjected to centrifugation at 5,000 rpm for 5 min. This centrifugation step effectively separated the solid components of the mixture from the liquid supernatant. The supernatant, containing the soluble proteins, was collected for further analysis.
To quantify the protein content in the supernatant, the method developed by *Lowry et al. (1951)* was employed. This method relies on a colorimetric assay to measure the protein concentration present in the sample.

### Photosynthetic parameters

To estimate the photosynthetic pigments, including chlorophylls (Chl a, Chl b, and TChl.) and carotenoids, the methods described by *Arnon (1949)* and *Davis (1979)* were followed. A spectrophotometer (Hitachi-220; Hitachi, Tokyo, Japan) was used for the measurements. The formulas used for calculations were:

$$Chl.\ a\ (mg\ g^{-1}\ f.wt.) = [12.7(OD663) - 2.69(OD645)] \times \frac{V}{1000} \times W$$

$$Chl.\ b\ (mg\ g^{-1}\ f.wt.) = [22.9(OD645) - 4.68(OD663)] \times \frac{V}{1000} \times W$$

$$Total\ chl.\ (mg\ g^{-1}\ f.wt.) = [20.2(OD645) - 8.02\ (OD663)] \times \frac{V}{1000} \times W$$

$$Carotenoids\ (mg\ g^{-1}\ f.wt.) = [12.7(OD480) - 0.114\ (OD663)] - 0.638\ (OD645)/2500$$

### Total antioxidant activity

For the measurement of total antioxidant activity, a dried leaf sample weighing 0.5 g was placed in a test tube. To facilitate the extraction of antioxidants from the leaf tissue, 20 mL

of a 0.45% salt solution was added to the test tube. The sample was then subjected to heating in a water bath at 40 °C for a duration of 20 min. After the heating process, the test tube was centrifuged at 3,000 rpm for 30 min, enabling the separation of the supernatant from the solid residue. The supernatant, which contained the extracted antioxidants, was carefully separated and stored at −20 °C until further analysis. To measure the total antioxidant activity, the FTC (Ferric Thiocyanate) method described by *Rahmat et al. (2003)* was employed. This method involves assessing the ability of the antioxidants to inhibit lipid peroxidation by reacting with ferric ions.

## Anatomical parameters

To examine the anatomy of the root, stem, and leaf, the largest plant from each replicate was selected. For leaf anatomy, a 2 cm section was obtained from the leaf base of fully mature and sun-exposed leaves. For stem anatomy, a section was taken from the base of the internode of the main stem. Similarly, for root anatomy, a section was obtained from tap root near the junction of the root and shoot. The collected plant material was fixed using a formaldehyde acetic alcohol solution consisting of 10% formaldehyde, 5% acetic acid, 50% ethanol, and 35% distilled water. The plant material was immersed in the fixative solution for 48 h, followed by transfer to an acetic alcohol solution containing 25% acetic acid and 75% ethanol for long-term storage. To prepare the sections for microscopic analysis, free-hand sections were made from the fixed plant material. These sections underwent a series of dehydration steps using ethanol. For staining, the sections were subjected to the standard safranin and fast green double-staining technique, as outlined by *Ruzin (1999)*. Measurements of the sections were taken using a light microscope (Nikon SE Anti-Mould, Nikon, Tokyo, Japan) equipped with an ocular micrometer that was calibrated using a stage micrometer. Micrographs of the stained sections were captured using a digital camera (Nikon FDX-35) mounted on a stereomicroscope (Nikon 104, Japan).

## Statistical analysis

The morphological, physiological, and anatomical trait data were subjected to statistical analysis using a One-way analysis of variance (ANOVA) in a complete randomized design with ten replicates. Mean values were compared using the least significant difference (LSD) test at a significance level of 5%. The statistical analysis was conducted using the Minitab software package (version 17.1.0; Pennsylvania State University, State College, PA, USA). To examine the relationships between the different morphological, physiological, and anatomical traits and the soil physicochemical parameters of the collection sites, Principal Component Analysis (PCA) was conducted. The analysis was carried out using the R-studio software, and the data were plotted to visualize the patterns and associations. Furthermore, heatmaps were constructed using the pheatmap package in R-studio. These heatmaps were used to cluster the selected groups based on (i) soil physicochemical attributes and morphophysiological parameters, (ii) soil physicochemical attributes and root anatomy, (iii) soil physicochemical attributes and stem anatomy, and (iv) soil

physicochemical attributes and leaf anatomy. The heatmaps provide a visual representation of the relationships and similarities among the different variables.

## RESULTS

### Soil physicochemical characteristics

The soil in most of the habitats was sandy (Table 2). The loamy soil was observed in five habitats RYK (near the wasteland), KHP (near waste deposit), VEH (near the roadside), JHG (along rice field) and LYH (wheat field) whereas loamy sand was observed in two habitats such as MUL (along river Chenab) and RJP (near M5 motorway). Clayey loam was seen in DGK habitat (along railway track). The soil electrical conductivity ranged from 0.76 to 6.73 $dSm^{-1}$, the maximum value of soil ECe was recorded at RYK (near the wasteland) and KHP (near waste deposit) sites and the minimum was observed at SDK (along barren land) and RJP (near M5 motorway). Habitats like water channel (LAP), along Chenab river (MUL) and near GT road (JHG) showed exceptionally highly level of soil ECe than rest of the populations. Most of the habitat comprised of alkaline pH, ranging from 6.2 to 8.9. The acidic pH was observed only in one habitat RYK (near the wasteland). The soil organic matter (OM) varied from 0.21 to 0.56%. The maximum organic matter was noted in soil of Chenab river (MUL) and the minimum was measured in soil of roadside population (VEH). The soil saturation percentage (SP) ranged from 15 to 42%. The maximum saturation percentage was observed in soil of wheat filed (LYH) population. It was the minimum in soil of water canal (LAP) and GT road (JHG) populations. The soil Phosphate concentration varied from 1.6 mg $Kg^{-1}$ in the SDK habitat to 3.6 mg $Kg^{-1}$ in the LYH habitat. The nitrate content in the LYH habitat exhibited the highest value, while the MUZ habitat recorded the lowest value. The soil chloride ion ($Cl^-$) reached its maximum (567.8 mg $Kg^{-1}$) in the RYK habitat, while the minimum (72.1 mg $Kg^{-1}$) was observed in both the DGK and MUL habitats. The soil calcium ion ($Ca^{2+}$) concentration ranged from 54.2 to 156.1 mg $Kg^{-1}$. The RYK habitat showed the highest soil calcium concentration, while the SDK habitat exhibited the lowest concentration. The soil sodium ion ($Na^+$) ranged between 54.2 and 398.9 mg $Kg^{-1}$, with the RYK population having the highest value and the SDK habitat recording the lowest. The maximum soil potassium ion ($K^+$) concentration was observed in the MUL and LYH habitats, while the minimum was found in the SDK habitat.

### Growth characteristics

Plant height was the maximum (56.5 cm) in BWP population and the minimum (16.3 cm) in FSD population (Fig. 2 and Table 3). The maximum shoot length (44.7 cm) was recorded in BWP population while the minimum (11.3 cm) of this parameter was noted in FSD population. Three populations, KHP, VEH and SAR showed maximum shoot fresh (11.5 g $plant^{-1}$) and dry weight (5.8 g $plant^{-1}$), while population FSD had least shoot fresh (3.0 g $plant^{-1}$) and dry weight (1.2 g $plant^{-1}$). Root length was the maximum (11.5 cm) in BWP and the minimum (4.5 cm) in VEH population. Four populations namely RYK, KHP, LAP and SAR showed maximum root fresh weight (1.5 g $plant^{-1}$), while the population RJP exhibited low value of dry weight (0.4 g $plant^{-1}$). Population RYK showed

**Table 3 Growth and physiological attributes of differently adapted populations of star weed (*Parthenium hysterophorus* L.) collected from Punjab province.**

| Ecological regions | Near wasteland | | | Along water channel | | | | Along roadside | | | | Near agriculture field | | | | | |
|---|---|---|---|---|---|---|---|---|---|---|---|---|---|---|---|---|---|
| Collection sites | RYK | SDK | KHP | BWP | LAP | AHP | MUL | VEH | DGK | RJP | JHG | MUZ | SAR | FSD | LYH | LSD | F-value |
| **Growth attributes** | | | | | | | | | | | | | | | | | |
| Plant height (cm) | 37.0$^d$ | 46.0$^b$ | 40.3$^c$ | 56.5$^a$ | 45.0$^b$ | 31.0$^e$ | 39.5$^c$ | 41.2$^c$ | 37.3$^d$ | 20.3$^f$ | 37.7$^d$ | 36.3$^d$ | 43.0$^c$ | 16.3$^g$ | 30.0$^e$ | 11.6 | 72.6*** |
| Shoot length (cm) | 30.0$^d$ | 40.3$^b$ | 31.0$^d$ | 44.7$^a$ | 35.0$^c$ | 26.0$^e$ | 33.0$^{cd}$ | 36.3$^c$ | 30.4$^d$ | 16.3$^f$ | 28.0$^d$ | 30.0$^d$ | 33.0$^{cd}$ | 11.3$^g$ | 24.0$^e$ | 4.5 | 19.4*** |
| Shoot fresh weight (g plant$^{-1}$) | 6.3$^d$ | 8.2$^{bc}$ | 11.5$^a$ | 9.4$^b$ | 8.2$^{bc}$ | 5.4$^{de}$ | 7.7$^c$ | 11.0$^a$ | 4.5$^e$ | 4.5$^e$ | 9.4$^b$ | 4.7$^e$ | 11.5$^a$ | 3.0$^f$ | 4.4$^{de}$ | 2.2 | 14.9*** |
| Shoot dry weight (g plant$^{-1}$) | 3.1$^c$ | 4.1$^b$ | 5.8$^a$ | 4.7$^b$ | 4.1$^b$ | 2.7$^{cd}$ | 3.9$^b$ | 5.8$^a$ | 2.2$^d$ | 2.0$^d$ | 4.7$^b$ | 2.3$^d$ | 5.8$^a$ | 1.2$^e$ | 2.0$^{cd}$ | 1.4 | 11.8** |
| Root length (cm) | 8.0$^b$ | 6.0$^c$ | 8.0$^b$ | 11.5$^a$ | 10.0$^{ab}$ | 7.7$^b$ | 5.7$^c$ | 4.5$^d$ | 7.3$^{bc}$ | 7.7$^b$ | 10.0$^{ab}$ | 6.0$^c$ | 10.0$^{ab}$ | 6.0$^c$ | 7.2$^b$ | 1.8 | 31.7*** |
| Root fresh weight (g plant$^{-1}$) | 1.5$^a$ | 0.5$^{bc}$ | 1.5$^a$ | 0.7$^b$ | 1.5$^a$ | 0.6$^c$ | 0.7$^b$ | 0.7$^b$ | 0.7$^b$ | 0.4$^d$ | 0.6$^c$ | 0.8$^b$ | 1.5$^a$ | 0.5$^{bc}$ | 0.5$^c$ | 1.0 | 68.8*** |
| Root dry weight (g plant$^{-1}$) | 1.2$^a$ | 0.3$^e$ | 1.0$^{ab}$ | 0.2$^f$ | 1.0$^{ab}$ | 0.2$^f$ | 0.3$^e$ | 0.4$^d$ | 0.5$^c$ | 0.2$^f$ | 0.3$^e$ | 0.5$^c$ | 1.0$^{ab}$ | 0.3$^e$ | 0.2$^f$ | 0.5 | 86.1*** |
| Leaf number (per branch) | 29.5$^a$ | 18.5$^d$ | 25.5$^b$ | 17.0$^d$ | 19.0$^d$ | 10.5$^{ef}$ | 14.0$^e$ | 22.0$^c$ | 15.5$^e$ | 14.5$^e$ | 18.5$^d$ | 11.0$^{ef}$ | 20.5$^c$ | 9.0$^f$ | 9.5$^{ef}$ | 4.3 | 25.7*** |
| Leaf area (cm$^2$) | 14.9$^k$ | 53.4$^c$ | 19.2$^j$ | 65.5$^a$ | 38.4$^e$ | 39.2$^e$ | 23.9$^i$ | 65.4$^a$ | 38.0$^e$ | 16.6$^h$ | 27.6$^g$ | 47.9$^d$ | 59.3$^b$ | 33.7$^f$ | 37.2$^e$ | 9.8 | 8.3** |
| **Physiological attributes** | | | | | | | | | | | | | | | | | |
| Total soluble protein (µg g$^{-1}$ d.wt.) | 47.9$^a$ | 26.4$^f$ | 21.7$^g$ | 41.9$^b$ | 20.8$^g$ | 23.1$^g$ | 22.8$^g$ | 9.4$^i$ | 32.0$^d$ | 29.8$^e$ | 35.7$^c$ | 19.3$^h$ | 36.3$^c$ | 24.7$^g$ | 22.1$^g$ | 6.7 | 33.3*** |
| Proline (µmol g$^{-1}$ dwt.) | 8.8$^c$ | 8.8$^c$ | 7.0$^d$ | 19.8$^a$ | 5.9$^e$ | 1.6$^g$ | 3.5$^f$ | 3.6$^f$ | 6.7$^{de}$ | 10.4$^b$ | 10.8$^b$ | 8.9$^c$ | 7.9$^c$ | 8.5$^c$ | 1.6$^g$ | 9.1 | 39.1*** |
| Glycine betaine (µmol g$^{-1}$ dwt.) | 3.6$^b$ | 3.8$^b$ | 2.6$^c$ | 10.2$^a$ | 2.2$^c$ | 2.5$^c$ | 2.4$^c$ | 2.1$^c$ | 2.4$^c$ | 3.1$^b$ | 2.4$^c$ | 2.6$^c$ | 3.9$^b$ | 1.9$^d$ | 2.3$^c$ | 4.5 | 35.0*** |
| Chlorophyll a (mg g$^{-1}$ f. wt.) | 1.9$^c$ | 2.4$^a$ | 1.9$^c$ | 1.9$^c$ | 1.7$^{cd}$ | 2.2$^b$ | 1.3$^d$ | 1.7$^{cd}$ | 1.3$^d$ | 1.3$^d$ | 1.3$^d$ | 1.7$^{cd}$ | 2.2$^b$ | 1.3$^d$ | 2.0$^b$ | 1.3 | 52.8*** |
| Chlorophyll b (mg g$^{-1}$ f. wt.) | 0.3$^f$ | 2.0$^a$ | 0.7$^e$ | 1.8$^{ab}$ | 1.0$^d$ | 1.7$^{ab}$ | 1.8$^{ab}$ | 1.8$^{ab}$ | 0.8$^e$ | 1.5$^c$ | 2.0$^a$ | 2.0$^a$ | 1.3$^c$ | 2.0$^a$ | 1.6$^{ab}$ | 1.0 | 40.2*** |
| Total chlorophyll (mg g$^{-1}$ f. wt.) | 2.1$^f$ | 4.4$^a$ | 2.6$^d$ | 3.7$^{ab}$ | 2.7$^d$ | 3.9$^{ab}$ | 3.1$^c$ | 3.5$^b$ | 2.1$^f$ | 2.8$^e$ | 3.3$^b$ | 3.7$^{ab}$ | 3.5$^b$ | 3.3$^b$ | 3.7$^{ab}$ | 1.5 | 60.5*** |
| Carotenoids (mg g$^{-1}$ f. wt.) | 1.5$^d$ | 2.4$^b$ | 1.4$^d$ | 1.4$^d$ | 2.8$^a$ | 1.8$^c$ | 1.8$^c$ | 2.6$^{ab}$ | 1.8$^c$ | 1.7$^c$ | 1.9$^c$ | 1.0$^e$ | 2.5$^b$ | 1.7$^c$ | 1.6$^c$ | 1.1 | 18.8*** |
| Chlorophyll a/b | 6.3$^a$ | 1.2$^d$ | 2.7$^b$ | 1.0$^e$ | 1.7$^c$ | 1.2$^d$ | 0.7$^f$ | 0.9$^f$ | 1.6$^c$ | 0.8$^f$ | 0.6$^g$ | 0.8$^f$ | 0.3$^h$ | 0.6$^g$ | 1.0$^d$ | 0.5 | 73.1*** |
| Total Chlorophyll/carotenoid | 1.4$^e$ | 3.1$^{ab}$ | 1.0$^f$ | 2.6$^c$ | 1.3$^e$ | 0.9$^g$ | 1.7$^d$ | 0.3$^h$ | 1.1$^f$ | 1.6$^d$ | 1.7$^d$ | 3.7$^a$ | 1.4$^e$ | 1.9$^d$ | 0.7$^g$ | 0.9 | 89.2*** |
| Antioxidant activity (%) | 5.0$^d$ | 5.4$^d$ | 4.2$^e$ | 5.2$^d$ | 3.5$^f$ | 6.5$^c$ | 9.9$^a$ | 9.9$^a$ | 9.9$^a$ | 6.1$^c$ | 9.3$^{ab}$ | 5.7$^d$ | 6.2$^c$ | 7.8$^{bc}$ | 6.4$^c$ | 3.3 | 36.4*** |

Notes:

Abbreviations are given at start of manuscript. Means shearing similar letters in each row are statistically not significant.

\*\*Significant at $P < 0.01$; \*\*\*significant at $P < 0.001$.

Abbreviations are given as footnote of Figs. 7 and 8.

the maximum dry weight (1.2 g plant$^{-1}$) and populations BWP, AHP, RJP and LYH possessed the minimum dry weight (0.2 g plant$^{-1}$). The maximum number of leaves (29.5) were recorded in RYK population, while their minimum value (9.0) was observed in FSD population. Two populations, BWP (65.3 cm$^2$) and VEH (65.4 cm$^2$) showed the maximum value of leaf area, while the minimum (14.9 cm$^2$) of that parameter was measured in RYK population.

## Physiological characteristics

The population from RYK exhibited the highest total soluble protein content (47.9 µg g$^{-1}$ d.wt.), while the population from VEH had the lowest (9.4 µg g$^{-1}$ d.wt.) (Table 3). Population BWP showed the maximum proline content (19.8 µmol g$^{-1}$ d.wt.), whereas populations AHP and LYH possessed the minimum (1.6 µmol g$^{-1}$ d.wt.). Glycine betaine content was highest in the BWP population (10.2 µmol g$^{-1}$ d.wt.) and lowest in the FSD population (1.9 µmol g$^{-1}$ d.wt.). For chlorophyll a content, the SDK population had the highest value (2.4 mg g$^{-1}$ f. wt.), while populations MUL, DGK, RJP, JHG, and FSD had the lowest value (1.3 mg g$^{-1}$ f. wt.). Four populations, SDK, JHG, MUZ, and FSD

showed the highest chlorophyll b content (2.0 mg g$^{-1}$ f. wt.), whereas the RYK population showed the lowest value (0.3 mg g$^{-1}$ f. wt.). The SDK population had the maximum total chlorophyll content (4.4 mg g$^{-1}$ f. wt.), while the RYK and DGK populations had the minimum (2.1 mg g$^{-1}$ f. wt.). The LAP population had the highest carotenoid content (2.8 mg g$^{-1}$ f. wt.), and the MUZ population had the lowest (1.0 mg g$^{-1}$ f. wt.). The chlorophyll a/b ratio was highest in the RYK population (6.3) and lowest in the SAR population (0.3). The MUZ population had the maximum total chlorophyll/carotenoid ratio (3.7), whereas the VEH population had the minimum (0.3). Antioxidant activity was the maximum (9.9 %) in three populations, MUL, VEH and DGK, whereas it was the minimum (3.5%) in the LAP population.

## Anatomical characteristics

### Root anatomy

The maximum root area (400.4 µm$^2$) was recorded in two populations, SAR and RYK, whereas the minimum (259.1 µm$^2$) was in three populations such as KHP, BWP and FSD (Fig. 3 and Table 4). The population from MUL had the maximum epidermal thickness (31.4 µm), while the population from LYH had the minimum epidermal thickness (9.4 µm). Population RYK showed the maximum cortical thickness (94.2 µm), and population FSD had the smallest (31.4 µm). The largest cortical cells (41.1 µm$^2$) were recorded in RYK and KHP populations, whereas the smallest cells (7.4 µm$^2$) were seen in two populations, MUZ and SAR. Population BWP possessed the largest vascular bundles (121.3 µm$^2$) than rest of the populations. On the other hand, population MUL had smallest vascular bundles (55.0 µm$^2$). Three populations namely KHP, BWP and MUZ exhibited widened metaxylem vessels (15.7 µm$^2$), whereas the populations of VEH and SAR had the narrowest vessels (9.4 µm$^2$). Phloem area was the maximum (2.5 µm$^2$) in four populations, KHP, LAP, MUZ and FSD, but the minimum (0.5 µm$^2$) was recorded in BWP and JHG.

### Stem anatomy

The maximum value of stem area (440.4 µm$^2$) was observed in populations KHP and MUZ, while their minimum value (182.6 µm$^2$) was noted in JHG (Fig. 4 and Table 4). Epidermal thickness was the maximum (23.6 µm) in SAR and KHP, and the minimum (9.4 µm) in RYK, MUL, RJP andMUZ. Population KHP showed the highest cortical proportion (70.7 µm), whereas the populations of RYK and MUL had lowest region (18.8 µm) of that character. Cortical cells area was the maximum (14.1 µm$^2$) in population AHP, FSD and LYH, and the minimum (6.3 µm$^2$) was in BWP. Populations AHP and KHP showed largest vascular bundles (164.9 µm$^2$) as compared to other populations, while the populations of RYK, SDK and FSD represented smallest vascular regions (94.2 µm$^2$). The largest metaxylem vessels (18.8 µm$^2$) were recorded in KHP and MUL, and the smallest vessels (9.4 µm$^2$) were noted in the BWP, VEH, MUZ and LYH populations. Phloem area was the maximum (69.1 µm$^2$) in population LYH, and the minimum (14.1 µm$^2$) in SDK.

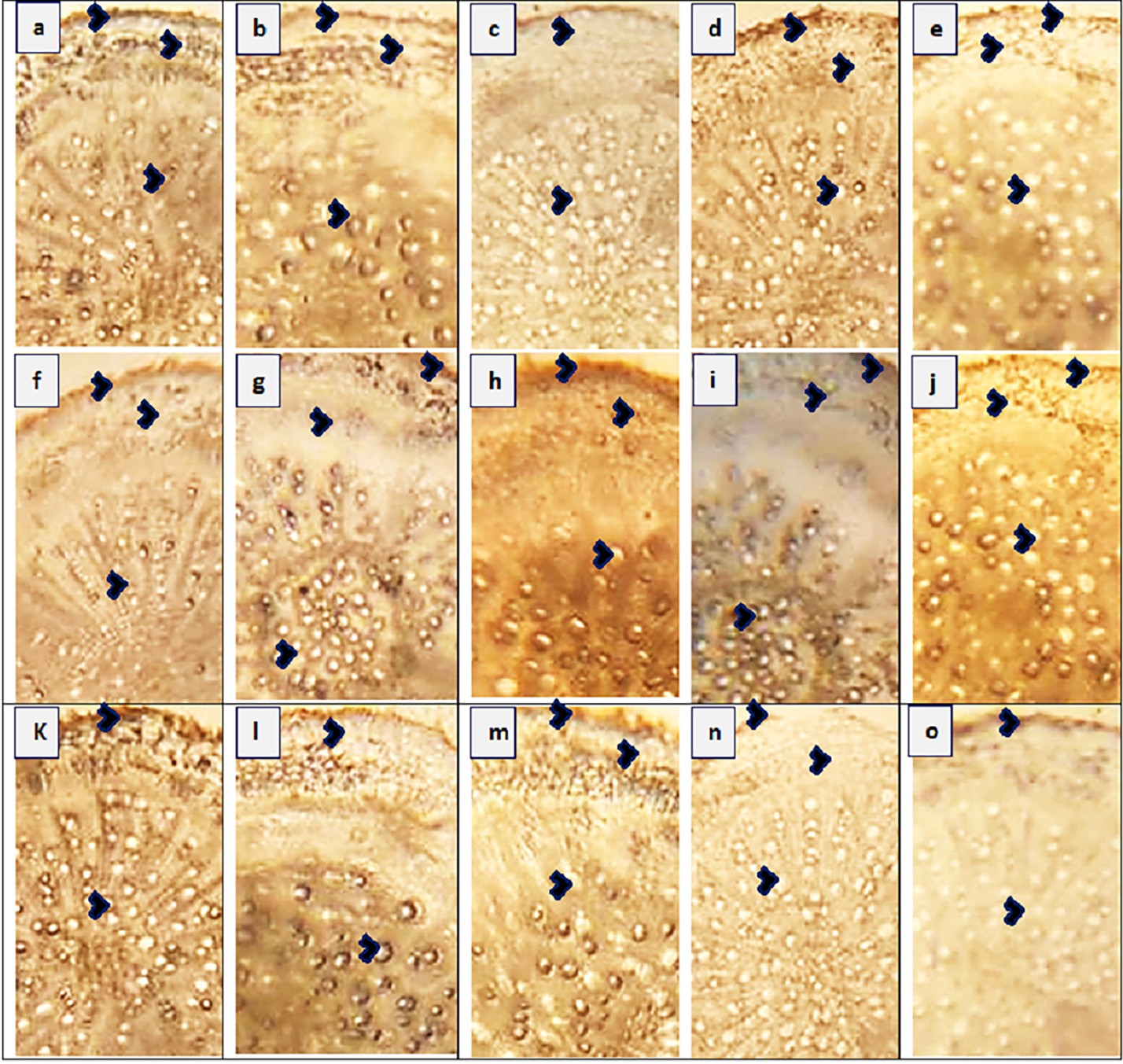

**Figure 3** **Root transvers sections of *Parthenium hysterophorus* L. populations collected from different ecological regions.** (A) RYK-Rahim Yar Khan. Thicker epidermis, enlarge cortical region and metaxylem vessels, (B) SDK-Sadiqabad. Reduced root cellular area, cortical thickness, metaxylem vessels and slightly crushed, (C) KHP-Khanpur. Reduced root area and epidermal thickness, enhanced metaxylem vessels (D) BWP-Bahawalpur. Greatly enhanced epidermal thickness, cortical thickness and metaxylem area, (E) LAP-Liaqatpur. Extraordinarily thicker epidermis, cortical region and metaxylem vessels, (F) AHP-Ahmadpur. Extraordinarily thick cortical region and enlarge metaxylem vessels, (G) MUL-Multan. Thick epidermis and cortical region, enhanced metaxylem area, (H) VEH-Vehari. Thicker epidermis, partially crushed cortical region and enlarge xylem vessels, (I) DGK-Dera Ghazi Khan. Thick epidermis and cortical region, reduced xylem vessels, (J) RJP-Rajanpur. Greatly reduced root cellular region and cortical thickness and metaxylem area, (K) JHG-Jhang. Reduced root area, cortical region and metaxylem vessels, (L) MUZ-Muzaffargarh. Reduced cortical thickness and partially crushed cortical region, (M) SAR-Sargodha. Thick epidermis, enlarge metaxylem vessels and cortical region, (N) FSD-Faisalabad. Thick cortical region and reduced xylem vessels, (O) LYH-Layyah. Reduced cortical region and metaxylem area.

**Table 4 Anatomical characteristics of differently adapted populations of star weed (*Parthenium hysterophorus* L.) collected from Punjab province.**

| Ecological regions | Near wasteland | | | Along water channel | | | | Along roadside | | | | Near agriculture field | | | | | |
|---|---|---|---|---|---|---|---|---|---|---|---|---|---|---|---|---|---|
| Collection sites | RYK | SDK | KHP | BWP | LAP | AHP | MUL | VEH | DGK | RJP | JHG | MUZ | SAR | FSD | LYH | LSD | F-ratio |
| **Root anatomy** | | | | | | | | | | | | | | | | | |
| Root area (μm²) | 400.4$^a$ | 282.6$^e$ | 259.1$^f$ | 259.1$^f$ | 306.2$^d$ | 304.6$^d$ | 306.2$^d$ | 306.2$^d$ | 353.3$^b$ | 329.7$^c$ | 282.6$^h$ | 329.7$^c$ | 400.4$^a$ | 259.1$^g$ | 353.3$^b$ | 10.9 | 66.5*** |
| Epidermal thickness (μm) | 18.8$^c$ | 17.3$^c$ | 12.6$^{de}$ | 18.8$^c$ | 22.0$^b$ | 17.3$^c$ | 31.4$^a$ | 17.3$^c$ | 14.1$^d$ | 15.7$^d$ | 22.0$^b$ | 15.7$^d$ | 22.0$^b$ | 14.1$^d$ | 9.4$^e$ | 5.4 | 47.3*** |
| Cortical thickness (μm) | 94.2$^a$ | 37.7$^e$ | 51.8$^d$ | 45.5$^d$ | 55.0$^c$ | 37.7$^e$ | 55.0$^c$ | 65.9$^b$ | 55.0$^c$ | 67.5$^b$ | 55.0$^c$ | 36.1$^e$ | 55.0$^c$ | 31.4$^f$ | 65.9$^b$ | 8.9 | 98.6*** |
| Cortical cell area (μm²) | 14.1$^a$ | 9.4$^c$ | 14.1$^a$ | 11.0$^b$ | 9.4$^c$ | 9.4$^c$ | 9.4$^c$ | 11.0$^b$ | 9.4$^c$ | 9.4$^c$ | 9.4$^c$ | 7.4$^d$ | 7.4$^d$ | 9.4$^c$ | 9.4$^c$ | 2.5 | 86.4*** |
| Vascular bundle areas (μm²) | 70.7$^c$ | 94.2$^b$ | 70.7$^c$ | 121.3$^a$ | 69.1$^c$ | 94.2$^b$ | 55.0$^e$ | 70.7$^c$ | 65.9$^d$ | 67.5$^b$ | 70.7$^c$ | 70.7$^c$ | 67.5$^d$ | 70.7$^c$ | 69.1$^c$ | 20.3 | 72.4*** |
| Metaxylem area (μm²) | 12.6$^c$ | 11.0$^d$ | 15.7$^a$ | 15.7$^a$ | 12.6$^c$ | 11.0$^d$ | 12.6$^c$ | 9.4$^e$ | 12.6$^c$ | 11.0$^d$ | 14.1$^b$ | 15.7$^a$ | 9.4$^e$ | 12.6$^c$ | 14.1$^b$ | 4.3 | 85.8*** |
| Phloem area (μm²) | 1.0$^c$ | 1.8$^b$ | 2.5$^a$ | 0.5$^d$ | 2.5$^a$ | 1.9$^b$ | 1.8$^b$ | 1.0$^c$ | 1.0$^c$ | 1.9$^b$ | 0.5$^d$ | 2.5$^a$ | 1.7$^b$ | 2.5$^a$ | 1.9$^b$ | 1.1 | 19.9*** |
| **Stem anatomy** | | | | | | | | | | | | | | | | | 71.6*** |
| Stem area (μm²) | 229.1$^g$ | 282.6$^e$ | 440.4$^a$ | 259.1$^f$ | 290.2$^d$ | 290.6$^d$ | 290.2$^d$ | 290.2$^d$ | 343.3$^b$ | 300.7$^c$ | 182.6$^h$ | 440.4$^a$ | 259.1$^f$ | 300.7$^c$ | 340.3$^b$ | 32.2 | 35.6*** |
| Epidermal thickness (μm) | 9.4$^d$ | 14.1$^c$ | 23.6$^a$ | 14.1$^c$ | 14.1$^c$ | 18.8$^b$ | 9.4$^d$ | 18.8$^b$ | 14.1$^c$ | 9.4$^d$ | 14.1$^c$ | 9.4$^d$ | 23.6$^a$ | 14.1$^c$ | 14.1$^c$ | 6.5 | 21.4*** |
| Cortical thickness (μm) | 18.8$^h$ | 23.6$^g$ | 70.7$^a$ | 33.0$^f$ | 55.0$^c$ | 47.1$^d$ | 18.8$^h$ | 47.1$^d$ | 67.5$^{ab}$ | 47.1$^d$ | 47.1$^d$ | 39.3$^e$ | 59.7$^b$ | 47.1$^d$ | 47.1$^d$ | 12.4 | 37.6*** |
| Cortical cell area (μm²) | 12.6$^b$ | 9.4$^c$ | 11.9$^b$ | 6.3$^d$ | 14.1$^a$ | 9.4$^c$ | 9.4$^c$ | 9.4$^c$ | 11.0$^b$ | 9.4$^c$ | 11.0$^b$ | 9.4$^c$ | 9.4$^c$ | 14.1$^a$ | 14.1$^a$ | 3.1 | 19.5*** |
| Vascular bundle area (μm²) | 94.2$^h$ | 94.2$^h$ | 164.9$^h$ | 131.9$^e$ | 108.3$^g$ | 164.9$^a$ | 146.0$^c$ | 117.8$^f$ | 146.0$^c$ | 149.2$^b$ | 128.7$^e$ | 117.8$^f$ | 133.5$^d$ | 94.2$^h$ | 133.5$^d$ | 12.7 | 49.4*** |
| Metaxylem area (μm²) | 12.6$^{bc}$ | 15.7$^b$ | 17.3$^a$ | 9.4$^c$ | 14.1$^b$ | 14.1$^b$ | 18.8$^a$ | 9.4$^c$ | 14.1$^b$ | 14.1$^b$ | 14.1$^b$ | 9.4$^c$ | 14.1$^b$ | 14.1$^b$ | 9.4$^c$ | 4.4 | 87.3*** |
| Phloem area (μm²) | 20.4$^f$ | 14.1$^g$ | 36.1$^e$ | 40.8$^d$ | 47.1$^c$ | 48.7$^c$ | 45.5$^c$ | 47.1$^c$ | 58.1$^b$ | 47.1$^c$ | 47.1$^c$ | 42.4$^d$ | 47.1$^c$ | 58.1$^b$ | 69.1$^a$ | 15.8 | 52.3*** |
| **Leaf anatomy** | | | | | | | | | | | | | | | | | 57.8*** |
| Midrib thickness (μm) | 379.9$^c$ | 420.8$^a$ | 376.8$^c$ | 337.6$^d$ | 329.7$^e$ | 235.5$^d$ | 329.7$^e$ | 389.4$^b$ | 376.8$^c$ | 329.7$^e$ | 329.7$^e$ | 329.7$^e$ | 282.6$^f$ | 235.5$^g$ | 282.6$^f$ | 12.9 | 18.5*** |
| Lamina thickness (μm) | 22.0$^c$ | 14.1$^e$ | 18.8$^d$ | 14.1$^e$ | 22.0$^c$ | 17.3$^d$ | 18.8$^d$ | 38.1$^a$ | 17.3$^d$ | 11.0$^f$ | 14.1$^e$ | 14.1$^e$ | 28.3$^b$ | 26.7$^{bc}$ | 14.1$^e$ | 6.4 | 73.8*** |
| Epidermal thickness (μm) | 18.8$^b$ | 15.7$^c$ | 23.6$^a$ | 14.1$^d$ | 23.6$^a$ | 23.6$^a$ | 10.6$^e$ | 15.7$^c$ | 14.1$^c$ | 12.6$^{de}$ | 17.3$^b$ | 17.3$^b$ | 15.7$^c$ | 18.8$^b$ | 15.7$^c$ | 3.3 | 89.4*** |
| Cortical thickness (μm) | 117.8$^h$ | 106.8$^i$ | 141.3$^e$ | 180.6$^{ab}$ | 139.7$^f$ | 117.8$^h$ | 150.7$^d$ | 185.3$^a$ | 153.9$^d$ | 158.6$^c$ | 127.2$^g$ | 122.5$^g$ | 139.7$^f$ | 100.1$^j$ | 119.3$^h$ | 17.6 | 36.3*** |
| Cortical cell area (μm²) | 14.1$^b$ | 14.1$^b$ | 15.7$^b$ | 15.7$^b$ | 12.6$^c$ | 11.0$^c$ | 15.7$^b$ | 18.8$^a$ | 11.0$^c$ | 14.1$^b$ | 14.1$^b$ | 18.8$^a$ | 14.1$^b$ | 9.4$^d$ | 9.4$^d$ | 4.4 | 72.7*** |
| Vascular bundle area (μm²) | 47.1$^f$ | 117.8$^a$ | 92.6$^c$ | 83.2$^d$ | 70.7$^e$ | 69.1$^e$ | 69.1$^e$ | 70.7$^e$ | 83.2$^d$ | 97.3$^b$ | 69.1$^e$ | 92.6$^c$ | 70.7$^e$ | 70.7$^e$ | 83.2$^d$ | 12.5 | 19.8*** |
| Metaxylem area (μm²) | 18.8$^e$ | 37.7$^a$ | 29.8$^b$ | 10.9$^g$ | 17.3$^e$ | 16.2$^{ef}$ | 14.8$^f$ | 22.0$^d$ | 22.0$^d$ | 20.4$^e$ | 25.1$^c$ | 17.3$^e$ | 18.8$^e$ | 14.1$^f$ | 22.0$^d$ | 8.0 | 14.5** |
| Phloem area (μm²) | 15.7$^c$ | 23.6$^b$ | 20.4$^{bc}$ | 16.0$^c$ | 14.1$^c$ | 11.8$^d$ | 16.0$^c$ | 20.4$^{bc}$ | 11.8$^d$ | 22.0$^{bc}$ | 23.6$^b$ | 20.4$^{bc}$ | 28.3$^a$ | 15.7$^c$ | 20.4$^{bc}$ | 5.2 | 11.3** |

**Notes:**
Abbreviations are given at the start of manuscript. Means shearing similar letters in each row are statistically not significant.
**Significant at *P* < 0.01; ***significant at *P* < 0.001.
Abbreviations are given as footnote of Figs. 7 and 8.

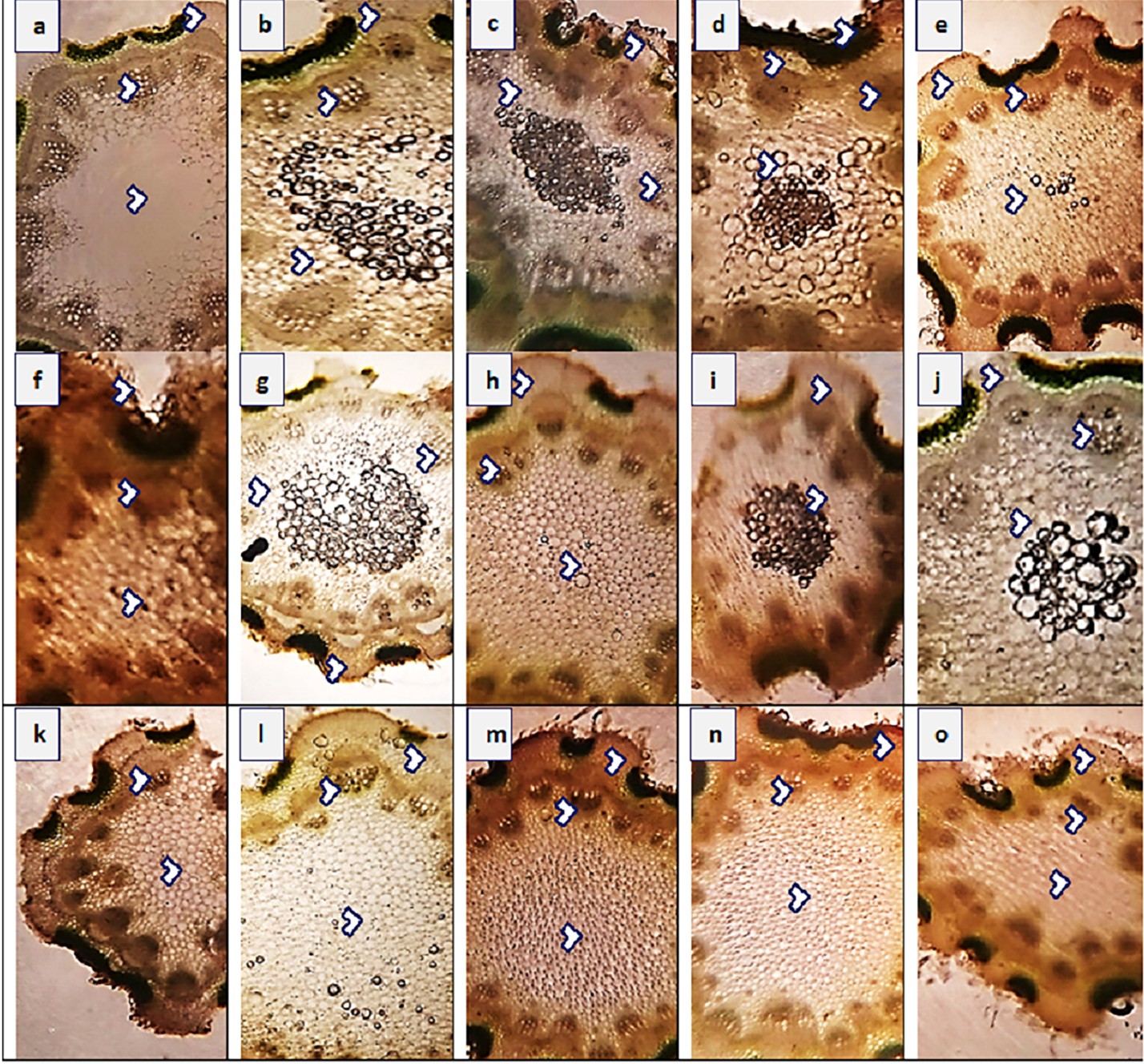

**Figure 4 Stem transvers sections of *Parthenium hysterophorus* L. populations collected from different ecological regions.** Description: (A) RYK-Rahim Yar Khan. Thicker epidermis, enlarge cortical region and vascular bundles, (B) SDK-Sadiqabad. Reduced stem cellular area, cortex thickness, metaxylem vessels and vascular bundle area, (C) KHP-khanpur. Enlarge stem area, enhanced cortical and epidermal thickness, sparse hairiness on surface (D) BWP-Bahawalpur. Greatly enhanced epidermal thickness, cortical and pith thickness, and vascular bundle area, (E) LAP-Liaqatpur. Extraordinarily thick cortical region, vascular and pith region, thick surface pubescence, (F) AHP-Ahmadpur. Extraordinary, reduced stem area, pith region and enlarge surface hairs, (G) MUL-Multan. Thick cortical region reduced vascular bundles and enhanced pith area, (H) VEH-Vehari. Thicker epidermis, partially crushed cortical region and reduced pith and vascular region, (I) DGK-DG Khan. Thicker cortical region enhanced vascular bundles and pith region, (J) RJP-Rajanpur. Greatly reduced stem area, pith thickness and vascular bundle area, (K) JHG-Jhang. Reduced stem area, vascular region and pith region, (L) MUZ-Muzaffargarh. Enhanced cortical thickness, vascular region and pith area, (M) SAR-Sargodha. Thicker epidermis, enlarge vascular bundles and pith region, (N) FSD-Faisalabad. Thick cortical region, surface hairiness, enlarge vascular bundles and xylem vessels, (O) LYH-Layyah. Reduced stem area, pith thickness and vascular area.

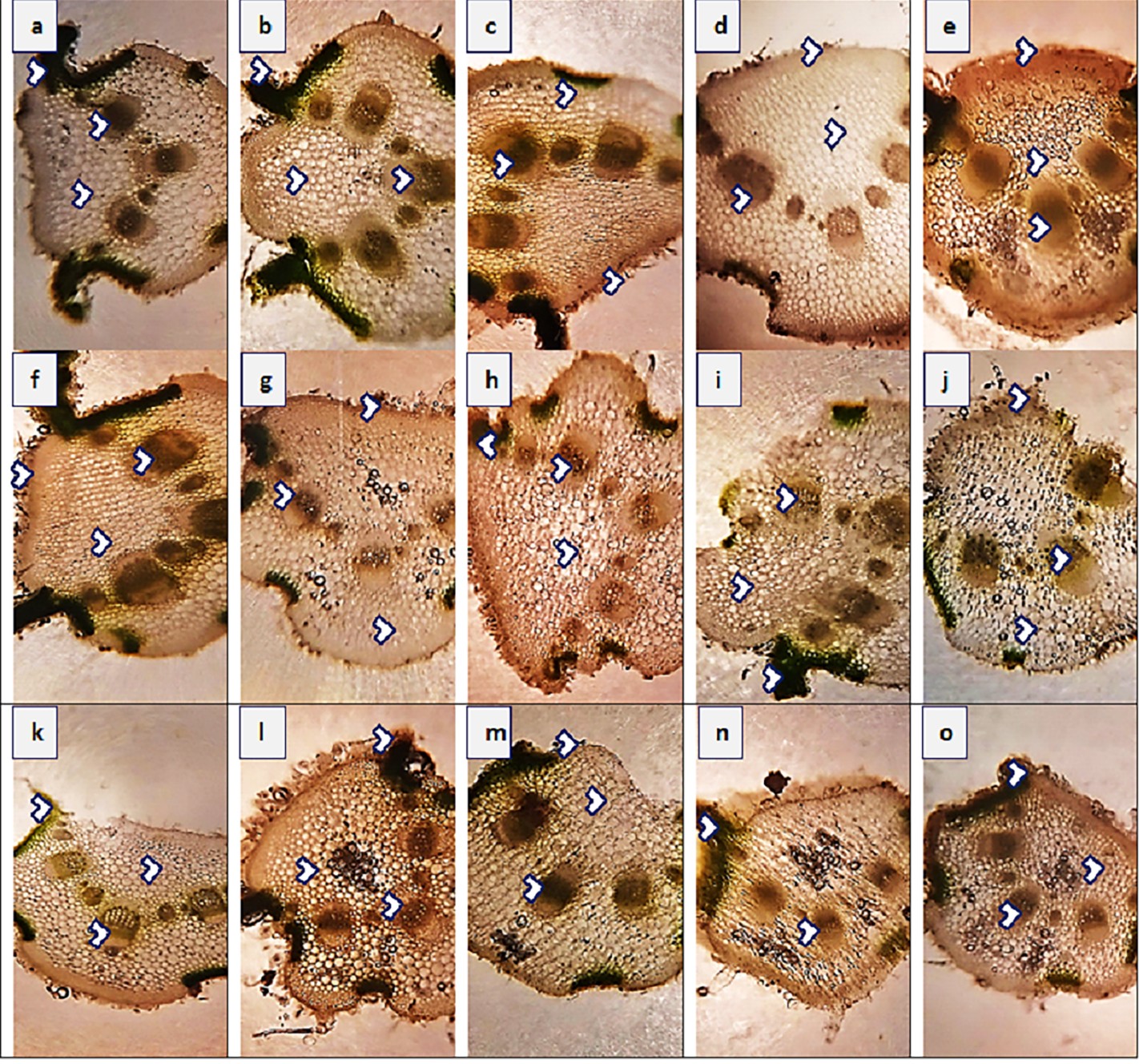

**Figure 5 Leaf transvers sections of *Parthenium hysterophorus* L. populations collected from different ecological regions.** Description: (A) RYK-Rahim Yar Khan. Thicker lamina, enlarge proportion of cortical region and reduced vascular bundles, (B) SDK-Sadiqabad. Thick leaf in terms of midrib and lamina thickness, enhanced cortex thickness and vascular region, (C) KHP-khanpur. Reduced leaf thickness, enlarge cortical region and vascular bundle area, (D) BWP-Bahawalpur. Greatly enhanced epidermal thickness, lamina thickness, cortical thickness and vascular area, (E) LAP-Liaqatpur. Extraordinarily thick leaf, cortical region and vascular bundles, (F) AHP-Ahmadpur. Extraordinarily thick cortical region, surface hairiness and reduced vascular bundles, (G) MUL-Multan. Reduced lamina thickness and epidermal thickness enhanced cortical region and vascular area, (H) VEH-Vehari. Thicker leaf, epidermis, enhanced cortical region and vascular bundle area, (I) DGK-DG Khan. Sparse surface hairiness, Thick cortical region, enhanced vascular region, (J) RJP-Rajanpur. Greatly reduced leaf thickness, cortical thickness and enlarged vascular bundle area, (K) JHG-Jhang. Thick leaf area, vascular bundles and cortical region, (L) MUZ-Muzaffargarh. Reduced lamina, cortical thickness and large vascular bundles, (M). SAR-Sargodha. Thick epidermis, enlarge vascular bundles and cortical region, (N) FSD-Faisalabad. Reduced leaf area, thick cortical region and reduced vascular bundles, (O) LYH-Layyah. Enhanced surface hairiness, thickness of cortical region and vascular bundles.

### Leaf anatomy

Leaf thickness greatly varied in all populations of *P. hysterophorus* (Fig. 5 and Table 4). Midrib thickness was the maximum (420.8 µm) in SDK, and the minimum (235.5 µm) in FSD population. The maximum value of lamina thickness (38.1 µm) was observed in population VEH, while the minimum value (11.0 µm) was observed in RJP. Thicker epidermis (23.6 µm) was measured in three populations, KHP, LAP and AHP, whereas the thinner (10.6 µm) of this parameter was noted in MUL. Enhanced cortical region (185.3 µm) was observed in VEH, and their reduced (100.1 µm) was in FSD. The population from roadside habitats (VEH) exhibited the largest cortical cells (18.8 µm$^2$), while the populations from FSD and LYH had the smallest cortical cells (9.4 µm$^2$). The vascular bundle area was highest (117.8 µm$^2$) in the SDK population, whereas the RYK population had the lowest vascular bundle area (47.1 µm$^2$). Among the populations, SDK had the largest metaxylem vessels (37.7 µm$^2$), while BWP had the smallest (10.9 µm$^2$). The phloem area was greatest (28.3) in the SAR population but was minimal (11.8 µm$^2$) in the DGK and AHP populations.

## MULTIVARIATE ANALYSIS

### Principal component analysis (PCA)

Principal component analysis (PCA1) exhibited 27.4% and 21.2% (48.6%) variability among morpho-physiological and soil physicochemical characteristics of *P. hysterophorus*. The Chl a, TChl/Car, TChl, RDW, RFW, SFW and GB showed strong influence of soil NO$_3$, SP, PO$_4$ and pH, whereas Chl b, Chl a/b, TSP, SDW, SL, RL and LA represented least influence of soil OM (Fig. 6A). Principal component analysis revealed significant influence of soil physiochemical characters on anatomical traits of species. PCA2 represented the variability of 33.2% and 18.4% (51.6%) among root anatomy and soil physicochemical attributes, as the CCA showed close influence of soil Ca$^{2+}$, ECe, Cl$^-$ and Na$^+$, while MA, RA, CT, EpT, PhA and VBA had least influence of soil OM (Fig. 6B). PCA3 indicated 36.5% and 18.9% (55.4%) variations between stem anatomy and soil parameters, for example the MA represented very close influence of soil K and NO$_3$, whereas the CCA showed with soil ECe and VBA with soil pH (Fig. 6C). PCA4 exhibited 28.8% and 19.9% (58.8%) variability amid leaf anatomy and soil attributes, as the LMT, MrT and CCA showed strong influence of soil NO$_3$, K$^+$, SP and PO$_4$, while the EpT with soil Ca$^{2+}$ and Cl$^-$, and the VBA and PhA with soil pH (Fig. 6D).

### Clustered heatmaps

Heatmap between soil physicochemical characters and morpho-physiological attributes exhibited six major clusters (Fig. 7A). In the first cluster, soil attribute, OM form cluster with Ca and Car content. The second cluster indicated the clustering of soil ECe, Cl and Na with LN, RFW and RDW. In the third cluster, RL form cluster with chlb, Tchl, TSP and GB. The fourth group showed clustering of soil attributes K, NO3, SP and PO4. The fifth cluster exhibited the clustering of LA and soil pH, and the sixth cluster showed the clustering of Chl a/b, Chla and TChl/Car. The seventh cluster indicated the clustering of SDW, SFW, PH and SL. Heatmap between root anatomical characteristics and soil

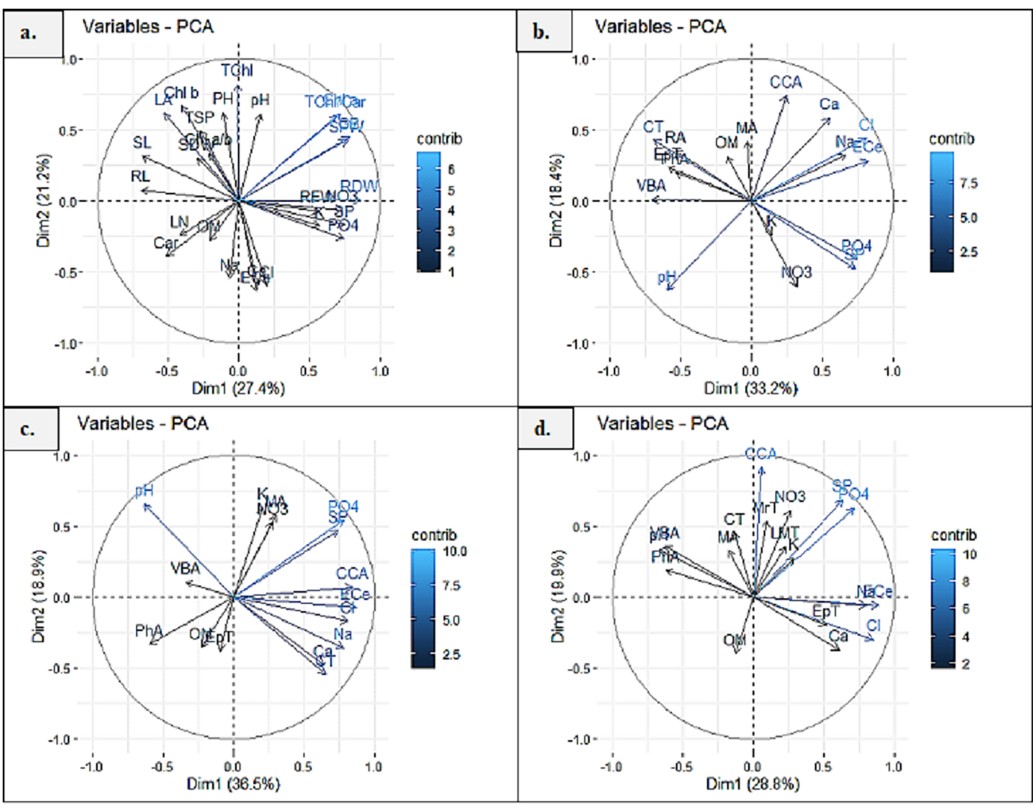

**Figure 6 Principal component analysis (PCA) showing influence of soil physicochemical characteristics on (A) growth and physiological features, (B) root anatomy, (C) stem anatomy, (D) leaf anatomy of *Parthenium hysterophorus* collected from Punjab province.** RYK-Rahim Yar Khan, SDK-Sadiq abad, KHP-Khanpur BWP Bahawalpur, LAP-Liaqatpur, AHP-Ahmadpur, MUL-Multan, VEH-Vehari, MUZ-Muzaffargarh, SAR-Sargodha, FSD-Faisalabad, DGK-Dera Ghazi Khan, RJP-Rajanpur, JHG-Jhang, LYH-Layyah. Soil: ECe-electrical conductivity, pH-soil pH, OM-organic matter, SP-saturation percentage, PO₄-phosphate, NO₃-nitrate, Cl-chloride, Ca-calcium, Na-sodium, K-potassium. Physiology: TSP-total soluble proteins, GB-glycine betaine, Chl a-chlorophyll a, Chl b-chlorophyll b, TChl-total chlorophyll, Car-carotenoids, Chl a/b-chlorophyll a/b ratio, TChl/Car-total chlorophyll/ carotenoid ratio. Morphology: PH-plant height, SFW-shoot fresh weight, SDW-shoot dry weight, LN-leaf number, RL-root length, RFW-root fresh weight, RDW-root dry weight, LA-leaf area, SL-shoot length. Root anatomy: RA-root area, EpT-epidermal thickness, CT-cortical thickness, CCA-cortical cell area, VBA-vasculer bundle area, MA-metaxyelm area, PhA-phloem area. Stem anatomy: SA-stem area, EpT-epidermal thickness, CT-cortical thickness, CCA-cortical cell area, VBA-vascular bundle area, MA-metaxyelm area, PhA-phloem area. Leaf anatomy: MrT-midrib thickness, LMT-lamina thickness, EpT-epidermal thickness, CT-cortical thickness, CCA-cortical cell area, VBA-vascular bundle area, MA-metaxylem area, PhA-phloem area.

attributes indicated four major clusters (Fig. 7B). The first cluster indicates the clustering of OM and MA. In the second cluster, soil pH form cluster with RA, CT, PhA, EpT and VBA. In the third cluster, soil attributes like NO3, K, SP and PO4 form clustering. In the fourth cluster, soil Ca, Na, ECe and Cl showed clustering with CCA.

The heatmap between soil physicochemical attributes and stem anatomical features exhibited three clusters (Fig. 7C). In the first cluster, soil pH and OM form clustering with VBA, EpT and PhA. In the second cluster, soil ECe, Cl, Na and Ca show clustering with CCA and CT. The third cluster indicated the clustering of K, PO4, SP and NO₃ with MA.

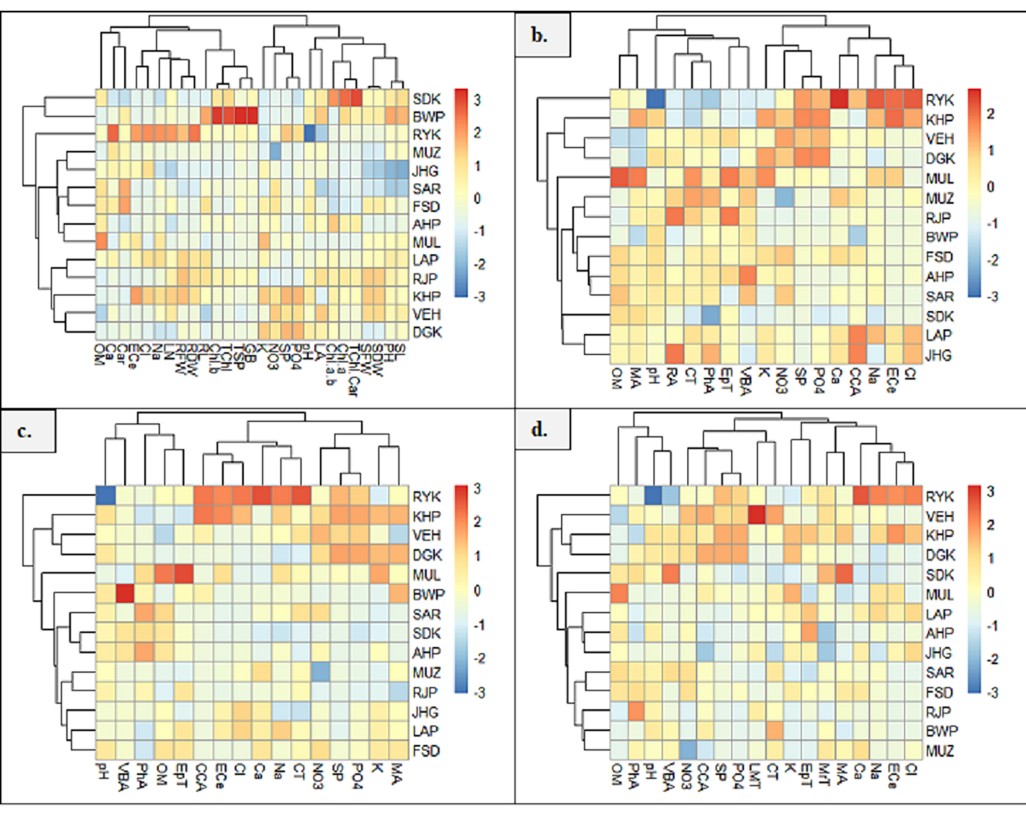

**Figure 7 Heatmap showing association of soil physiochemical characteristics on (A) growth and physiological characteristics, (B) root, (C) stem, and (D) leaf anatomical features of *Parthenium hysterophorus* collected from the Punjab province.** RYK-Rahim Yar Khan, SDK-Sadiq abad, KHP-Khanpur BWP-Bahawalpur, LAP-Liaqatpur, AHP-Ahmadpur, MUL-Multan, VEH-Vehari, MUZ-Muzaffargarh, SAR-Sargodha, FSD-Faisalabad, DGK-Dera Ghazi Khan, RJP-Rajanpur, JHG-Jhang, LYH-Layyah. Soil: ECe-electrical conductivity, pH-soil pH, OM-organic matter, SP-saturation percentage, $PO_4$-phosphate, $NO_3$-nitrate, Cl-chloride, Ca-calcium, Na-sodium, K-potassium. Physiology: TSP-total soluble proteins, GB-glycine betaine, Chl a-chlorophyll a, Chl b-chlorophyll b, TChl-total chlorophyll, Car-carotenoids, Chl a/b-chlorophyll a/b ratio, TChl/Car-total chlorophyll/carotenoid ratio. Morphology: PH-plant height, SFW-shoot fresh weight, SDW-shoot dry weight, LN-leaf number, RL-root length, RFW-root fresh weight, RDW-root dry weight, LA-leaf area, SL-shoot length. Root anatomy: RA-root area, EpT-epidermal thickness, CT-cortical thickness, CCA-cortical cell area, VBA-vasculer bundle area, MA-metaxyelm area, PhA-phloem area. Stem anatomy: SA-stem area, EpT-epidermal thickness, CT-cortical thickness, CCA-cortical cell area, VBA-vascular bundle area, MA-metaxyelm area, PhA-phloem area. Leaf anatomy: MrT-midrib thickness, LMT-lamina thickness, EpT-epidermal thickness, CT-cortical thickness, CCA-cortical cell area, VBA-vascular bundle area, MA-metaxylem area, PhA-phloem area               

Heatmap between soil physicochemical attributes and leaf anatomical features exhibited four clusters (Fig. 7D). In the first cluster, soil OM and pH form cluster with PhA and VBA, whereas in the second cluster, soil NO3, SP and PO4 form cluster with LMT and CCA. The third cluster indicates the clustering of EPT and CT, while the fourth cluster showing clustering of soil ECe, NA, Cl and Ca with MrT and MA.

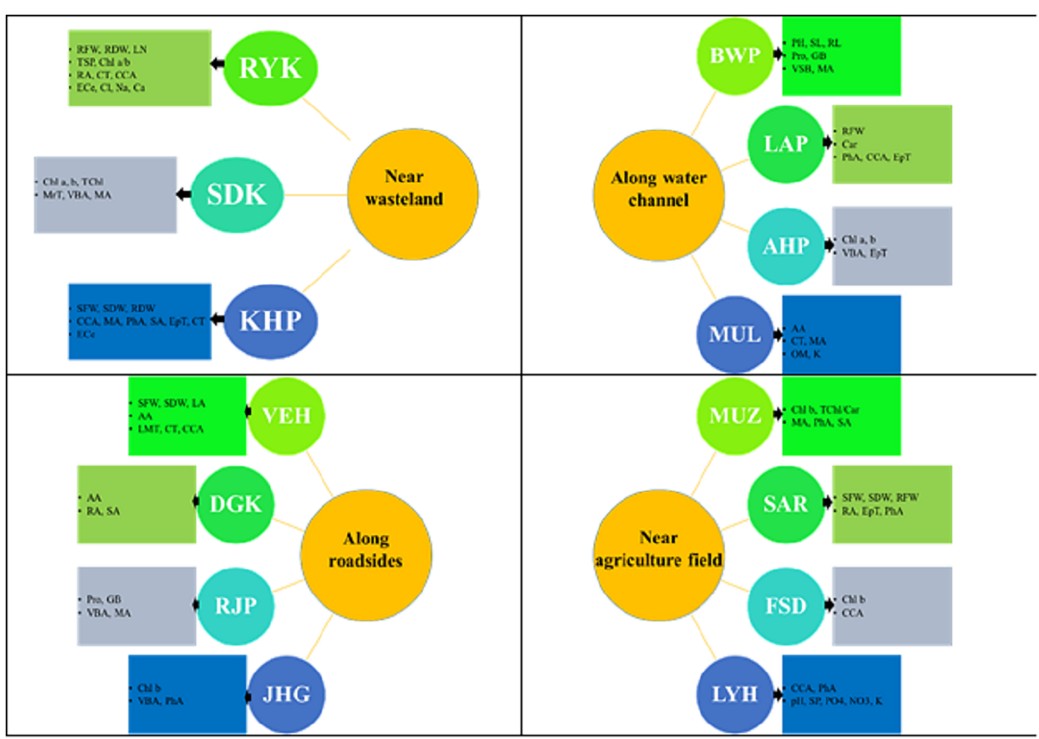

**Figure 8** Prominent structural and functional adaptations in different populations of *Parthenium hysterophorus* collected from the Punjab province. RYK-Rahim Yar Khan, SDK-Sadiq abad, KHP-Khanpur, BWP-Bahawalpur, LAP-Liaqatpur, AHP-Ahmadpur, MUL-Multan, VEH-Vehari, MUZ-Muzaffargarh, SAR-Sargodha, FSD-Faisalabad, DGK-Dera Ghazi Khan, RJP-Rajanpur, JHG-Jhang, LYH-Layyah. Soil: ECe-electrical conductivity, pH-soil pH, OM-organic matter, SP-saturation percentage, $PO_4$-phosphate, $NO_3$-nitrate, Cl-chloride, Ca-calcium, Na-sodium, K-potassium. Physiology: TSP-total soluble proteins, GB-glycine betaine, Chl a-chlorophyll a, Chl b-chlorophyll b, TChl-total chlorophyll, Car-carotenoids, Chl a/b-chlorophyll a/b ratio, TChl/Car-total chlorophyll/carotenoid ratio. Morphology: PH-plant height, SFW-shoot fresh weight, SDW-shoot dry weight, LN-leaf number, RL-root length, RFW-root fresh weight, RDW-root dry weight, LA-leaf area, SL-shoot length. Root anatomy: RA-root area, EpT-epidermal thickness, CT-cortical thickness, CCA-cortical cell area, VBA-vasculer bundle area, MA-metaxyelm area, PhA-phloem area. Stem anatomy: SA-stem area, EpT-epidermal thickness, CT-cortical thickness, CCA-cortical cell area, VBA-vascular bundle area, MA-metaxyelm area, PhA-phloem area. Leaf anatomy: MrT-midrib thickness, LMT-lamina thickness, EpT-epidermal thickness, CT-cortical thickness, CCA-cortical cell area, VBA-vascular bundle area, MA-metaxylem area, PhA-phloem area.

## DISCUSSION

A summary of specific adaptive strategies of differently adapted populations of *Parthenium hysterophorus* collected from different regions of Punjab province are highlighted in Fig. 8. The evaluation of morpho-anatomical and physio-biochemical adaptive markers is crucial for understanding the underlying mechanisms of adaptation in differently adapted populations to multiple stresses (*Hameed, Ashraf & Naz, 2011*; *Nawaz et al., 2023*). In the face of severe drought conditions or physiological drought induced by other environmental stresses, water conservation becomes a primary strategy (*Sun et al., 2018*). In water-scarce conditions, water conservation in plants is achieved through mechanisms such as water storage in parenchymatous tissues like pith and cortex (*Alvarez, Rocha &*
*Machado, 2008*; *Iqbal, Hameed & Ahmad, 2021*), efficient water translocation facilitated by widening of vessels, and reduction of water loss through the presence of mechanical tissues and a thick cuticle on the surface of plant organs (*Micco & Aronne, 2012*). Herein, we tested the strength of adaptation and the extent of these adaptations in plant survival, different populations of *P. hysterophorus* were sampled from a wide range of habitats. It was hypothesized that the invasive success of *P. hysterophorus* in diverse habitats is influenced by its phenotypic plasticity, allowing it to adapt to a wide range of environmental conditions.

The investigation revealed significant variations in morphological characteristics among the populations of *P. hysterophorus*, which can be attributed to the diverse environmental conditions in which these populations were originally adapted. Under diverse conditions where the *P. hysterophorus*, populations were collected, the genetically fixed characteristics of each population were expressed, reflecting their adaptation to their respective habitats (*Mojica et al., 2012*; *Paccard, Vance & Willi, 2013*). The population from the BWP site, which is located along a water channel with relatively soft soil texture, exhibited the maximum growth (Table 3). This type of habitat seems to be more favorable for the growth and development of *P. hysterophorus*, as reported for other hydrophytes (*Qadir et al., 2008*; *Hasanuzzaman et al., 2014*). The compactness of the soil directly influenced the growth and propagation of the species, with habitats consisting of compact soil showing shorter plants, such as the FSD and VEH populations. Similar findings were reported by *Hamza & Anderson (2005)*, who observed shorter stature plants in compact soil. Biomass production, both in roots and shoots, is a reliable criterion for assessing tolerance potential of a species (*Khosroshahi et al., 2014*). The RYK and KHP populations demonstrated good overall growth response, indicating their potential for stress tolerance. The SAR population also exhibited vigorous growth, suggesting its complete adaptation to its specific habitat. Root and shoot parameters, such as length, number, fresh and dry weights, have been previously associated with abiotic stresses like drought or physiological drought in other plant species (*Talukdar, 2013*; *Ye et al., 2015*). The RYK population displayed a high number of leaves per plant, although they were smaller in size. Having a large number of leaves can enhance a plant's photosynthetic efficiency (*Weraduwage et al., 2015*), while smaller leaves can increase water use efficiency by reducing transpiration rates (*Medranoa et al., 2015*). This adaptation is particularly important for survival in harsh saline desert conditions.

Chlorophyll pigments serve as sensitive indicators of the metabolic state under salt stress conditions (*Chattopadhyay et al., 2011*). In the present study, the least saline population SDK and moderately saline population KHP showed an increase in chlorophyll *a*, chlorophyll *b*, total chlorophyll, and carotenoid content. Similar findings have been reported by *Amirjani (2011)* on rice *Triticum aestivum* L. and *Sarabi et al. (2017)* on melon (*Cucumis melo* L.). They noted an augmentation in photosynthesis-related parameters under moderate salinity levels, but a decline was observed under high salinity conditions. Conversely, the highly saline population RYK exhibited lower amounts of chlorophyll pigments and carotenoids. This decrease in pigment content aligns with other studies that have reported a significant reduction in photosynthetic pigments under highly saline conditions, such as *López-Millán et al. (2009)* in *Lycopersicon esculentum*, *Peng, Kroneck &*

*Küpper (2013)* in *Elsholtzia splendens*, and *Sytar et al. (2013)* in various plant species. In the present study, the BWP population showed an increasing trend in organic osmolytes. The accumulation of osmolytes is an effective strategy employed by plants to endure prevailing, which serves as a defensive mechanism for plants to maintain turgor pressure and prevent tissue collapse due to desiccation (*Kholodova, Volkov & Kuznetsov, 2010*; *Sun et al., 2010*). Elevated levels of total antioxidant activity were observed in *P. hysterophorus* populations inhabiting roadside areas, such as VEH and DGK. These findings align with previous studies conducted by *Nadgorska-Socha, Ptasinska & Kita (2013)*, *Zemanova et al. (2013)*, and *Almohisen (2014)*, which demonstrated that dust pollution stimulates the production of various metabolites in plants. These metabolites play a crucial role in mitigating stress by activating the plants' defense systems (*Sharma & Dietz, 2006*).

The anatomical characteristics of plants have been recognized as highly responsive to climatic conditions (*Caemmerer & Evans, 2015*; *Iqbal, Hameed & Ahmad, 2021*). This adaptability enables plants to thrive and survive in challenging environment (*De Micco & Aronne, 2012*). The size of the root cross-sectional area is predominantly determined by the relative proportions of the cortical region and the vascular bundle area (Table 4). An expansion in root area not only enhances the capacity for water storage but also strengthens the mechanical integrity of the plant's soft tissues. This, in turn, facilitates the efficient transportation of water and minerals from the roots to the aerial parts of the plant, aided by physiological adjustments. The observed increase in root cross-sectional area indicates better growth in the population inhabiting waste land (RYK). Roots, being underground plant parts, are relatively less affected by environmental conditions compared to other plant organs (*Fitter & Hay, 2012*). Epidermis is an outermost protective layer of roots, and under harsh condition its strong friction of rhizospheric soil (*McKenzie et al., 2013*). In resulting, this may be damaged, mainly in grasses and herbs (*McCully, 1999*). *P. hysterophorus* showed a significant increase of this parameter in MUL population (along water channel). Thicker epidermal layers play vital role in resisting the friction of soil compaction as well as impede the excessive water and solute translocation inside root tissues (*Chimungu, Loades & Lynch, 2015*). The water storage parenchyma (cortex) and vascular region (metaxylem vessels and phloem) in the roots play a crucial role, especially during water deficit or saline conditions. These adaptations are particularly significant for the survival of arid zone species such as *P. hysterophorus* (*Hsiao & Xu, 2000*). A significantly increased storage parenchyma and vascular region has been observed in populations of KHP (along waste deposit) and MUZ (along agriculture field).

The plants growing in wastelands (KHP) demonstrated the highest values for the majority of stem anatomical characteristics, as shown in Table 4. These characteristics encompass dermal, vascular, and storage tissues, indicating favorable growth conditions and enhanced biomass production, as evidenced by the shoot fresh weight. These findings are consistent with previous studies conducted by *Engloner (2009)* and *Guo & Miao (2010)*. The presence of sclerified tissues in the stems is a notable adaptation to dry conditions (*Nikolova & Vassilev, 2011*). It was recorded in stems from almost all habitats, but in populations from roadsides (VEH and DGK), there was higher lignin deposition compared to the other populations. Under extreme dry and hot conditions, tissue sclerification is

beneficial for preventing from collapse of internally metabolically active tissues during desiccation (*da Cruz Maciel et al., 2015*; *Ahmad et al., 2016*).

In arid zone species like *P. hysterophorus*, the leaf blade plays a vital role as it needs to withstand harsh environmental conditions for the plant's survival. Among the studied populations, the plants from roadside habitats (VEH) exhibited the highest values for various leaf anatomical characteristics, including leaf thickness in terms of midrib and lamina thickness, as well as mechanical and storage tissues such as cortical thickness and its cells area. These adaptations are indicative of the plant's ability to protect the leaf blade from the challenging environmental conditions encountered in roadside habitats (*Ameer et al., 2023*). Three populations, namely KHP (near wasteland), AHP (near Punjab Barrage) and LAP (along agriculture field), possessed thick epidermis and spare surface hairiness. Both are effective for evapo-transpiration loss when population surviving in dry environmental condition (*Gonzáles et al., 2008*; *Sarwar et al., 2022*). Overall, these results suggest that phenotypic plasticity in structural and functional adaptations of *P. hysterophorus* contribute to its resilience, competitive ability, and allowing it to adapt to a wide range of environmental conditions, making it a successful and problematic invasive species.

## CONCLUSION

In conclusion, *P. hysterophorus* displays significant variations in both structural and functional attributes, enabling it to tolerate diverse environmental adversities. The wide distribution of this species can be attributed to its specific adaptations along environmental gradients. It exhibits a range of adaptations, including changes in growth parameters, microstructural features, and functional traits. These adaptations, such as enhanced biomass production, long and numerous roots, thicker epidermis, development of storage parenchyma tissues, lignification of cortical region and vascular bundles, and increased levels of organic osmolytes and antioxidants. Overall, the structural and functional adaptations of *P. hysterophorus* contribute to its resilience, competitive ability, and ability to colonize a wide range of habitats, making it a successful and problematic invasive species.

## ACKNOWLEDGEMENTS

The content of this manuscript has been derived from the MPhil thesis of the second author.

### Funding

The authors received no funding for this work.

### Competing Interests

Khawaja Shafique Ahmad is an Academic Editor for PeerJ. There is no competing interest related to this work.

## Author Contributions

- Ummar Iqbal conceived and designed the experiments, analyzed the data, prepared figures and/or tables, authored or reviewed drafts of the article, and approved the final draft.
- Zartasha Usman performed the experiments, prepared figures and/or tables, and approved the final draft.
- Akkasha Azam performed the experiments, prepared figures and/or tables, and approved the final draft.
- Hina Abbas analyzed the data, authored or reviewed drafts of the article, and approved the final draft.
- Ansar Mehmood analyzed the data, authored or reviewed drafts of the article, and approved the final draft.
- Khawaja Shafique Ahmad conceived and designed the experiments, authored or reviewed drafts of the article, and approved the final draft.

## Data Availability

Data for statistical analysis and codes used for the analysis are available in the Supplemental Files.

## Supplemental Information

Supplemental information for this article can be found online at http://dx.doi.org/10.7717/peerj.16609#supplemental-information.

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
