# Peer review of "Invasive success of star weed (Parthenium hysterophorus L.) through alteration in structural and functional peculiarities"

_PeerJ, doi:10.7717/peerj.16609_

## Round 0.1 · original submission · Major Revisions

Star weed (Parthenium hysterophorus L.) has special importance in agricultural and non-agricultural fields due to its impressive invasiveness potency. Each study that is conducted to understand how this plant is so successful in terms of filling all ecological niches is valuable. Therefore, this article can provide some important info to manage star weed populations. However, this article needs a major revision or rewriting.

Please edit your article in terms of linguistics.

Please rewrite your hypotheses and discuss them in an understandable way with more articles.

Please check all the suggestions of the reviewers, and correct your article according to their views.

If you do not accept any of these suggestions, you should explain the reason why you reject them.

**Language Note:** The Academic Editor has identified that the English language must be improved. PeerJ can provide language editing services - please contact us at copyediting@peerj.com for pricing (be sure to provide your manuscript number and title). Alternatively, you should make your own arrangements to improve the language quality and provide details in your response letter. – PeerJ Staff

Reviewer 1 ·

Basic reporting

• Language is average, poor at some points.
• Introduction portion need more contextual. Need to focus more on invasion success rather than other portions like promotion of sustainable agriculture
• The hypothesis set in line 70-77 is not well explained and justified in discussion portion
• Tables and figures are too difficult to understand

Experimental design

• Experimental design is ok but at some points it is confusing. Authors are requested to make it clear. Please see the reviewed manuscript in track change mode for detail.
• Research questions that were set are not clearly explained in the discussion section

Validity of the findings

Research findings in this paper are important to understand the invasion success of the notorious weed like Parthenium in various habitats. But the way of presentation is not clear to the reader.

Additional comments

Authors are advised to read some references related to Parthenium which will help enhance the quality of the manuscript -
Bajwa et al. 2016. What do we really know about alien plant invasion? A review of the invasion mechanism of one of the world’s….
Mao et al. 2021. A superweed in the making: adaptations of Parthenium hysterophorus to a changing climate. A review...

Annotated reviews are not available for download in order to protect the identity of reviewers who chose to remain anonymous.

·

Basic reporting

* No comment

Experimental design

* No comment

Validity of the findings

* No comment

Additional comments

In the article, it is understood that the environmental reasons (soil, habitat, etc.) underlying Parthenium hysterophorus L. being a successful invasive plant are associated with morphophysiological and some biochemical parameters. The data was statistically correlated with both PCA and heat maps. Data interpreted with an ecological approach can be a source for other research.
There are no deficiencies except for minor spelling errors (presented in the attached file).

Reviewer 3 ·

Basic reporting

No comment.

Experimental design

Comparison between plants grown under field conditions is very problematic, many things can change along the growth of the plant. Also, within field variation also exist and thus may affect the response of these plants even if they grow in the same location.

Validity of the findings

In my opinion, the data collected by the authors may only serve as preliminary results, leading to the test of each factor alone or in combination with other factors under controlled conditions. Thus, the authors may assess the true effect of each factor. At this point, there are too many factors and parameters and it is very hard to measure the effect of each one.
General conclusions drawn from this study are very fragile as they may change completely by changing only one factor.

Additional comments

The tables are suffering from an overload of data and it is very hard to read them, especially in Tables 3 and 4.
Specific comments
39: this paragraph should come after the paragraph on Parthenium (58-67).
61-62: also to the middle east.
86-88: if you marked 50, why did you use only 10, how did you select these 10 out of the 50 plants marked?
122: how did you extract the root from the soil? How can you be sure that you extracted the whole root from all soil types?
128-129: was leaf area measured for all leaves of each plant?
220: field and not filed.
351: what do you mean by controlled conditions? From what I understand, all samples were collected in the field.
374-375: was this study done on Parthenium hysterophorus?
393-395: this is only an assumption, further tests showing that water use efficiency of these plants and other physiological parameters are needed in order to prove it.

---

## Round 0.2 · Minor Revisions

Please erase the following references from your list because you did not cite them in your edited article

Safdar ME, Tanveer A, Khaliq A, Riaz MA. 2015. Yield losses in maize (Zea mays) infested with parthenium weed (Parthenium hysterophorus L.). Crop Protection, 70: 77−82.

Zulfiqar F, Casadesús A, Brockman H, Munné Bosch SM. 2019. An overview of plant-based natural biostimulants for sustainable horticulture with a particular focus on Moringa leaf extracts. Plant Science, 110194.

Additionally,
Use Fig. instead of Figure in your edited article.

Please write which leaves were selected to find the leaf area in your article.

Please remove all spaces before the sign of the degree.

---

## Round 0.3 · accepted · Accept

I would like to thank you for accepting of referees' suggestions and improving your article based on their suggestions. I think your article is ready to publish. We look forward to your next article.